# Systematic Exploration and Exploitation via a Markov Game with Impulse Control

## Abstract

Efficient reinforcement learning (RL) involves a trade-off between "exploitative" actions that maximise expected reward and "explorative" actions that lead to the visitation of "novel" states. To encourage exploration, existing methods proposed methods such as injecting stochasticity into action selection, implicit regularisation, and synthetic heuristic rewards. However, these techniques do not necessarily offer systematic approach for making this trade-off. Here we introduce **SE**lective **R**einforcement **E**xploration **N**etwork (SEREN), a plug-and-play framework that casts the exploration-exploitation trade-off as a Markov game between an RL agent – Exploiter, which purely exploits task-dependent rewards, and another RL agent – Switcher, which chooses at which states to activate a *pure exploration* policy that is trained to minimise system uncertainty and override Exploiter. Using a form of policies known as *impulse control*, Switcher is able to determine the best set of states to switch to the exploration policy while Exploiter is free to execute its actions everywhere else. We prove that the convergence of SEREN under linear regime, and show that it induces a natural schedule towards pure exploitation. Through extensive empirical studies in both discrete and continuous control benchmarks, we show that with minimal modification, SEREN can be readily combined with existing RL algorithms and yield performance improvement.

## 1 Introduction

Reinforcement learning (RL) is a framework that enables autonomous agents to learn complex behaviours through trial and error (Sutton & Barto, 2018). With the combination of neural-network based function approximations, RL has had notable successes in a number of practical domains such as robotics and games (Silver et al., 2016; Reed et al., 2022). In order to find the global optimal policy, the RL agent needs to trade off visiting states with known high rewards (exploitation) against going to (current) suboptimal states (exploration) during training, such that sufficient coverage of the state space is achieved. However, randomly perturbing actions for exploration is sample inefficient since it does not take into account the information acquired from previous experiences. In practice, this procedure exacerbates the sample complexity of the agent's learning of the optimal policy, despite theoretically grounded asymptotic convergence (Dabney et al., 2020).

Here we tackle the challenge of performing systematic and efficient exploration in RL. We propose a novel two-agent framework that disentangles exploration and exploitation into independent RL problems for more efficient learning of the respective optimal behaviours. **SE**lective **R**einforcement **E**xploration **N**etwork (SEREN) entails an interdependent interaction between an RL agent, Exploiter, whose goal is to maximise the current estimate of future task-dependent (extrinsic) rewards, and an additional RL agent, Switcher, whose goal is to explore so as to reduce system uncertainty across the state space. Furthermore, at any given state, Switcher has the power to override Exploiter and assume control of the system (at that state) to apply exploratory actions. Therefore, Switcher acts to reduce system uncertainty in subregions of the state space in which (high) system uncertainty exists. A key ingredient of the SEREN framework is the use of a form of policy known as *impulse control* (Øksendal & Sulem, 2007; Mguni et al., 2022) used by Switcher. This enables Switcher to quickly determine the appropriate points to activate its exploration policy to minimise system uncertainty.

By using a two-agent framework for independent learning of the exploitation and exploration policies, the competing individual goals of completing the task set by the environment versus exploration over the state space are decoupled and each delegated to an independent agent. This means that Exploiter pursues its task of maximising its objective by purely exploiting without trading-off rewards from the environment for exploration. Moreover, as Switcher itself is an RL agent, it learns to perform systematic and targeted arbitration between exploitative and exploratory actions, switching to exploration only where such actions produce a reduction (surpassing certain threshold, see Section 3) in the cummulative system uncertainty. We formally prove in Section 4 that an optimal schedule of exploration naturally emerges from SEREN without the need for heuristic exploration scheduling. SEREN is a flexible plug-and-play framework that can be easily integrated with existing RL algorithms. We instantiate SEREN on both value-based and policy-gradient RL agents and empirically evaluate on both discrete and continuous control benchmarks. We show that SEREN leads to systematic improvement over existing baselines on all presented tasks.

The intuition behind our framework is inspired by naturally occurring learning phenomena. A well-established hypothesis of animal decision-making is that animals exhibit information-seeking behaviour to reduce the internal estimates of the uncertainty of the environment (Gottlieb et al., 2013). Experimental evidence indicates the orthogonal encoding of information value and primary reward value in primate orbitofrontal cortex (OFC) for curiosity-based decision making (Blanchard et al., 2015), which coheres nicely with SEREN's framework of dual systems for independent learning of exploitative and exploratory behaviours (see further discussion in Section 7).

## 2    Preliminaries

In RL problems, an agent gradually learns to select actions for maximisation of its expected returns through interactions with the environment. The underlying maximisation problem is typically formalised as a Markov Decision Process (MDP): $\langle \mathcal{S}, \mathcal{A}, \mathcal{P}, \mathcal{R}, \gamma \rangle$, where $\mathcal{S} \subset \mathbb{R}^p$ is the set of states, $\mathcal{A} \subset \mathbb{R}^k$ is the action space, $\mathcal{P} : \mathcal{S} \times \mathcal{A} \times \mathcal{S} \to [0,1]$ is the transition function describing the environment dynamics, $\mathcal{R} : \mathcal{S} \times \mathcal{A} \to \mathbb{R}$ is the environmental reward function and the discounting factor $\gamma \in [0,1]$ specifies the degree to which the agent's future rewards are discounted over time (Sutton & Barto, 2018). The goal of an RL agent is to learn an optimal policy, $\pi^* : \mathcal{S} \to \mu(\mathcal{A})$, where $\mu(\mathcal{A})$ is a probability measure over $\mathcal{A}$, such that the expected value function is maximised for all $s \in \mathcal{S}$:

$$\pi^* = \arg\max_{\pi \in \Psi} v_\pi(s), v_\pi(s) = \mathbb{E}_\pi \left[ \sum_{\tau=0}^{\infty} \gamma^k \mathcal{R}_{t+\tau}(s_{t+\tau}, a_{t+\tau}) | S_t = s \right], \tag{1}$$

where $\mathcal{R}_t := \mathcal{R}(s_t, a_t)$ for all $t = 0, 1, \ldots, \forall s_t \in \mathcal{S}, \forall a_t \in \mathcal{A}$, $\Psi = \{\pi : \mathcal{S} \to \mu(\mathcal{A})\}$ is the policy space[1], and $S_t$ is the random variable of the state value at time $t$, given the policy $\pi$ and the transition dynamics $\mathcal{P}$.

## 3    SEREN: A Dual System for Exploitation and Exploration

SEREN consists of an RL agent, Exploiter and an *impulse control* (Mguni et al., 2022; 2021b;a) agent, Switcher. Switcher has the ability to transfer control of the system to the exploration policy and does so at a set of states it chooses. Switcher is trained to minimise the system uncertainty across the state space, hence determining the best set of states to transfer control to the exploration policy, whilst Exploiter is free to exploit everywhere else. The schematic illustration of the SEREN can be found in Figure 1.

As opposed to standard methods for exploration, we separate the exploration and exploitation into two separate agents. We do so by introducing another learned decision of when to switch from exploitative to explorative beahviour, and vice versa. To simplify learning, we consider two agents: Exploiter, which learns a standard exploitation policy by only maximising the *extrinsic reward*; and Switcher, which is trained to decide when to switch to explorative behaviour and learns a purely exploration policy by only maximising an uncertainty-based *intrinsic reward* (see equation 4). Thus, each agent operates on respective action spaces,

---

[1]Unless stated otherwise, we use $\mu(\cdot)$ to denote some probability measure over the specified space.

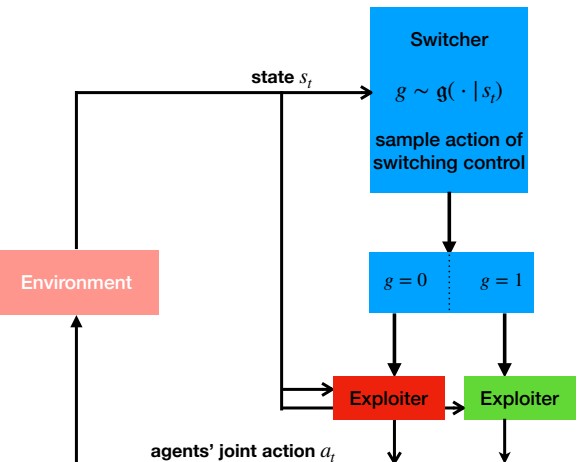

Figure 1: Schematic illustration of SEREN components and how SEREN interacts with the environment for action selection.

$\mathcal{A}^{\text{it}} \equiv \mathcal{A}$ and $\mathcal{A}^{\text{re}} \equiv \mathcal{A} \times \{0, 1\}$ (where $\mathcal{A}$ is the environment action space), with respective reward functions, $\mathcal{R}^{\text{it}}, \mathcal{R}^{\text{re}} : \mathcal{S} \times \mathcal{A}^{\text{it}} \times \mathcal{A}^{\text{re}} \to \mathbb{R}$ (see below for definition of $\mathcal{R}^{\text{re}}$)[2].

Formally, our framework is defined by a tuple $\mathcal{G} = \langle \mathcal{N}, \mathcal{S}, \mathcal{A}^{\text{it}}, \mathcal{A}^{\text{re}}, \overline{\mathcal{P}}, \mathcal{R}^{\text{it}}, \mathcal{R}^{\text{re}}, \gamma_{\text{it}}, \vec{\Gamma}_{\text{re}} \rangle$ where $\mathcal{N} = \{Exploiter, Switcher\}$ are the sets of agents, the transition probability, $\overline{\mathcal{P}} : \mathcal{S} \times \mathcal{A}^{\text{it}} \times \mathcal{A}^{\text{re}} \times \mathcal{S} \to [0, 1]$, which takes the current state and the actions of both agents as inputs, and the discounting factors, $\gamma_{\text{it}}$, $\vec{\Gamma}_{\text{re}} := (\gamma_{\text{re}}, \gamma_{\mathfrak{g}}) \in (0, 1) \times (0, 1)$, for $\pi^{\text{it}}$, $\pi^{\text{re}}$ and $\mathfrak{g}$, respectively (see below). Exploiter defines a Markov policy $\pi^{\text{it}} : \mathcal{S} \to \mathcal{A}^{\text{it}}$, which is contained in the set $\Psi^{\text{it}} \subseteq \Psi$. Switcher has two components: a Markov policy $\pi^{\text{re}} : \mathcal{S} \to \mathcal{A}$ from $\Psi^{\text{re}} \subseteq \Psi$, which determines the exploration action based on the measure of cumulative future uncertainty, and a discrete *switching* policy $\mathfrak{g} : \mathcal{S} \to \{0, 1\}$. By default, the joint system relies on the exploitation policy, $\pi^{\text{it}}$ to take actions. At each state Switcher makes a *binary decision* based on $\mathfrak{g}$ to decide whether to transfer control of the system to the exploration policy, $\pi^{\text{re}}$. We denote by $\{\tau_k\}_{k \geq 0}$ the timepoints at which the Switcher decides to activate the exploration policy, or simply the *intervention times*. The intervention times obey the expression $\tau_k = \inf\{t > \tau_{k-1} | s_t \in \mathcal{S}, \text{ s.t., } \mathfrak{g}(s_t) = 1\}$ and are therefore **rules** *that depend on the state*. Hence, by learning an optimal $\mathfrak{g}$, Switcher learns the best states to activate exploration. As we later explain, these intervention times are determined by a condition on the state which is easy to evaluate (see Prop. 2). For notational simplicity, we denote $\mathcal{R}^{\mathfrak{z}}_t = \mathcal{R}^{\mathfrak{z}}(s_t, a^{\text{it}}_t, a^{\text{re}}_t, g_t)$ for $\mathfrak{z} \in \{\text{it}, \text{re}\}$, where $g_t = \mathfrak{g}(s_t)$, $a^{\text{it/re}} \sim \pi^{\text{it/re}}(\cdot | s_t)$.

## 3.1 The Exploiter Objective

The goal of Exploiter is to (greedily) maximise its expected cumulative reward set by the environment. The objective that Exploiter seeks to maximise is:

$$v^{\text{it}}(s | \pi^{\text{it}}, \{\pi^{\text{re}}, \mathfrak{g}\}) = \mathbb{E}\left[\sum_{\tau=0}^{\infty} \gamma^{\tau}_{\text{it}} \mathcal{R}^{\text{it}}(s_{t+\tau}, a^{\text{it}}_{t+\tau}, a^{\text{re}}_{t+\tau}, g_{t+\tau}) | S_t = s\right], \forall s \in \mathcal{S}$$

$$\text{where } \mathcal{R}^{\text{it}}(s_t, a^{\text{it}}_t, a^{\text{re}}_t, g_t) = \mathcal{R}(s_t, a^{\text{it}}_t)(1 - g_t) + \mathcal{R}(s, a^{\text{re}}_t)g_t$$

(2)

where $a^{\text{it}}_t \sim \pi^{\text{it}}(\cdot | s_t)$ is the exploitative action, $a^{\text{re}}_t \sim \pi^{\text{re}}(\cdot | s_t)$ is the exploratory action at time $t$, $g_t$ is the binary output of the switcher (Section 3.3) that controls the execution of actions of Exploiter or Explorer. Therefore, the reward received by Exploiter is $\mathcal{R}(s_t, a^{\text{re}}_t)$ when $t = \tau_k$, $k = 1, 2, \dots$ i.e. whenever the Switcher activates the exploration policy and $\mathcal{R}(s_t, a^{\text{it}}_t)$ otherwise.

---

[2]Note that we use $\cdot^{\text{it}}$ and $\cdot^{\text{re}}$ to refer to the quantities associated with the Exploiter and Switcher, respectively.

Whenever Switcher decides to transfer control to the exploration policy, the exploration policy overrides the Exploiter and the integrated transition dynamics (given both the environment transition and the policy) are affected by only the exploration policy (while Exploiter influences the dynamics at all other times). The overall transition dynamics are therefore given by $\overline{\mathcal{P}}(s_{t+1}, a_t^{\mathrm{re}}, a_t^{\mathrm{it}}, g_t, s_t) := \mathcal{P}(s_{t+1}, a_t^{\mathrm{it}}, s_t)(1 - g_t) + \mathcal{P}(s_{t+1}, a_t^{\mathrm{re}}, s_t)g_t$.

## 3.2 The Exploration Policy

The actions selected by the exploration policy $\pi^{\mathrm{re}}$ are chosen so as to maximise the following objective function:

$$v^{\mathrm{re}}(s|\pi^{\mathrm{it}}, \{\pi^{\mathrm{re}}, \mathfrak{g}\}) = \mathbb{E}\left[\sum_{\tau=0}^{\infty} \gamma_{\mathrm{re}}^{\tau} \mathcal{R}^{\mathrm{re}}(s_{t+\tau}, a_{t+\tau}^{\mathrm{it}}, a_{t+\tau}^{\mathrm{re}}, g_{t+\tau})|S_t = s\right], \forall s \in \mathcal{S} \tag{3}$$

$$\text{where } \mathcal{R}^{\mathrm{re}}(s_t, a_t^{\mathrm{it}}, a_t^{\mathrm{re}}, g_t) := L(s_t, a_t^{\mathrm{re}})g_t + L(s_t, a_t^{\mathrm{it}})(1 - g_t)$$

where $L(s, a)$ is some measure of the system uncertainty about the state space which we specify in detail shortly, and is chosen to satisfy the property that $L \to 0$ as the system uncertainty decreases (see, e.g., Equation 4). Analogous to the reward function for Exploiter, the function $\mathcal{R}^{\mathrm{re}}$ is defined so that the received reward is $L(s_t, a_t^{\mathrm{re}})$ when $t = \tau_k$, $k = 0, 1, \ldots$ i.e. whenever the Switcher activates the exploration policy, and $L(s_t, a_t^{\mathrm{it}})$ otherwise.

We note that both $\mathcal{R}_t^{\mathrm{it}}$ and $\mathcal{R}_t^{\mathrm{re}}$ are evaluated at all timesteps, instead of independently when exploitation and exploration actions are respectively chosen, since extrinsic reward is received during exploratory behaviour and system uncertainty is also received during exploitative behaviour. Hence the joint system (SEREN) can maximally utilise the *reward* signals available (regardless whether exploitative or explorative behaviour is activated at each timestep) and learns to choose the optimal combination of exploitative and exploratory actions to maximise the respective objectives across all timepoints.

## 3.3 The Switching Mechanism

SEREN then utilises intrinsic rewards for choosing when to switch between exploitative and exploratory behaviour Schmidhuber (1991).

In principle, SEREN could accommodate various intrinsically constructed measures of uncertainty or novelty, possible choices include the model-based ensemble epistemic uncertainty in dynamics prediction (Chua et al., 2018), action-prediction errors (Pathak et al., 2017), and causally instructed novelty based on retrospective information (Yu et al., 2024), etc. Here we focus on the model-free instantiation of the proposed SEREN framework, hence we choose to employ a model-free estimate of uncertainty across the state space. In this case, we assume that the Exploiter employs an ensemble of neural networks as its value function. We quantify uncertainty over the state space using the non-parametric ensemble disagreement based on ensemble modelling of the value function of the Exploiter (Osband et al., 2016). In particular, for an ensemble of $M$ critic estimates of $\{Q_1, \ldots, Q_M\}$, we use the ensemble-based empirical estimate of uncertainty for any $(s, a) \in \mathcal{S} \times \mathcal{A}$:

$$L(s, a) = \frac{1}{M - 1} \sum_{m=1}^{M} (Q_m(s, a) - \mathcal{E}_Q(s, a))^2, \tag{4}$$

where $\mathcal{E}_Q(s, a) := \frac{1}{M} \sum_{m=1}^{M} Q_m(s, a)$ is the empirical mean of the ensemble predictions.

The goal of Switcher is to take actions that lead to states with high uncertainty, such that the system uncertainty over the state (-action) space is minimised. To induce Switcher to *selectively* choose when to switch to exploration, each switching activation incurs a fixed cost. These costs are quantified by the indicator function which is $\beta$ whenever an exploratory action is performed and 0 otherwise, where $\beta$ is a fixed positive constant. The presence of the *switching cost* ensures that the gain in measured uncertainty for performing exploratory actions to arrive at a given set of states is sufficiently high to merit forgoing rewards from taking exploitative actions in the corresponding states. Therefore to maximise this objective, Switcher must determine the sequence of points $\{\tau_k\}$ at which the benefit of performing an exploratory action

overcomes the cost of doing so. Accordingly at time $t \in 0, 1, \ldots$, Switcher seeks to maximise the following objective:

$$v^{\mathfrak{g}}(s|\pi^{\mathrm{it}}, \{\pi^{\mathrm{re}}, \mathfrak{g}\}) = \mathbb{E}\left[\sum_{\tau=0}^{\infty} \mathcal{R}^{\mathfrak{g}}(s_{t+\tau}, a_{t+\tau}^{\mathrm{it}}, a_{t+\tau}^{\mathrm{it}}, g_{t+\tau}; \beta)|S_t = s\right], \forall s \in \mathcal{S}$$

$$\text{where } \mathcal{R}^{\mathfrak{g}}(s_{t+\tau}, a_{t+\tau}^{\mathrm{it}}, a_{t+\tau}^{\mathrm{it}}, g_{t+\tau}; \beta) = \gamma_{\mathfrak{g}}^{\tau} L(s_{t+\tau}, a_{t+\tau}^{\mathrm{it}}) - g_{t+\tau}(L(s_{t+\tau}, a_{t+\tau}^{\mathrm{it}}) - L(s_{t+\tau}, a_{t+\tau}^{\mathrm{re}}) + \beta)$$

$$(5)$$

Therefore to maximise its objective, Switcher must determine the best set of states to reduce system uncertainty. Note that since $\mathcal{R}^{\mathrm{re}}$ depends on the uncertainty measure $L$ equation 4, which is in turn dependent on the rapidly changing value network, hence it has the non-stationary property that $\mathbb{E}[\mathcal{R}^{\mathrm{re}}(s)] \to 0$ as system uncertainty decreases across the state space. With low levels of uncertainty, the switching cost dominates so the Switcher does not intervene, leaving the Exploiter to take actions that deliver high rewards. This effectively pushes the Switcher out of the game as systemic uncertainty is reduced. This is precisely the behaviour that we seek as more information about the environment becomes known. We formally prove this property in Sec. 4 (see Prop. 2).

The pseudocode for off-policy training of SEREN is shown in Algorithm 1.

### 3.4   Relation to Other Exploratory Mechanisms

SEREN entails an exploration framework of great generality, such that many existing exploration models can be viewed as some degenerate form of SEREN. For instance, the classical $\epsilon$-greedy exploration can be interpreted as SEREN with a random switching mechanism and uniformly random exploration policy. Moreover, if we consider the case in which Switcher has the identical objective to Exploiter, consisting of task rewards with additive intrinsic rewards, then the model is equivalent to curiosity-based exploration (Schmidhuber, 1991; Pathak et al., 2017; Burda et al., 2018). We hope the framework of a dual system for exploration-exploitation tradeoff proposed in the current paper could inspire the development of more systematic exploration methods currently unthought of.

### 3.5   SEREN Training

As we show in Sec. 4, the learning processes for both agents converge to a stable solution. Note that since $\pi^{\mathrm{it}}$ and $\pi^{\mathrm{re}}$ share the same action space ($\mathcal{A}$), the Exploiter is trained off-policy using the data generated by both exploration and exploitation policies. The three policies, $\pi^{\mathrm{it}}$, $\pi^{\mathrm{re}}$, and $\mathfrak{g}$, maintain their independent replay buffers with respective actions and reward functions. We note that as the training progresses, the uncertainty inevitably decreases, yielding the reward function for the exploration policy training non-stationary. To counteract the non-stationarity of the reward structure in the learning of $\pi^{\mathrm{re}}$, the discounting factor $\gamma_{\mathrm{re}}$ is set to be appropriately lower (comparing to standard values) such that the agent still learns a policy that maximises future returns (instead of only relying on immediate uncertainty as in existing works in using intrinsic rewards), but at the same time reduces the negative effects caused by the distributional shift in the reward distribution.

## 4   Convergence & Optimality of SEREN

A key aspect of SEREN is the presence of two RL agents that each adapt their play according to the other's behaviour. This produces two concurrent learning processes each designed to fulfill distinct objectives. At a stable point of the learning processes Switcher minimises uncertainty about less explored states while Exploiter maximises the environment reward. Introducing simultaneous learners can occasionally lead to issues that prevent convergence to the stable point (Zinkevich et al., 2006).

We now show that $\mathcal{G}$ admits a stable point and that our method converges to it. In particular, we show that the joint system converges in its value functions for each agent. Additionally, we show that SEREN induces a natural schedule in which as the environment is explored, the number of switching operations (induced by Switcher) tends to 0. All proofs can be found in Appendix B.

We begin by stating a key result:

---

**Algorithm 1 SElective Reinforcement Exploration Network (SEREN)**

---

1: Given reward objective function for Switcher, uncertainty objective function $L(\cdot, \cdot)$, initialise independent Replay Buffers $\mathcal{B}^{\mathfrak{z}}$ for $\mathfrak{z} \in \{\text{it}, \text{re}, \mathfrak{g}\}$, switching cost $\beta$, episode length $T_{\text{eps}}$.
2: **for** $N_{episodes}$ **do**
3:      Reset state $s_0$
4:      **for** $t = 0, 1, \ldots, T_{\text{eps}}$ **do**
5:          Sample Exploiter action, $a_t^{\text{it}} \sim \pi^{\text{it}}(\cdot|s_t)$; Explorer action, $a_t^{\text{re}} \sim \pi^{\text{re}}(\cdot|s_t)$; Switcher action, $g_t \sim \mathfrak{g}(s_t)$,
6:          **if** $g_t = 0$ **then**
7:              Apply $a_t^{\text{it}}$ so $s_{t+1} \sim \mathcal{P}(\cdot|a_t^{\text{it}}, s_t)$,
8:          **else**
9:              Apply $a_t^{\text{re}}$ so $s_{t+1} \sim \mathcal{P}(\cdot|a_t^{\text{re}}, s_t)$,
10:          **end if**
11:          Compute rewards: $r_t^{\text{it}}$ (Equation 2), $r_t^{\text{re}}$ (Equation 3), and switcher reward $r_t^{\mathfrak{g}}$ (Equation 5).
12:          Store $(s_t, a_t^{\mathfrak{z}}, s_{t+1}, r_t^{\mathfrak{z}})$ in $\mathcal{B}^{\mathfrak{z}}$ for $\mathfrak{z} \in \{\text{it}, \text{re}, \mathfrak{g}\}$;
13:      **end for**
14:      **// Independent training of individual policies**
15:      Sample batches of $|B|$ transitions, $B^{\mathfrak{z}} = \{(s_b^{\mathfrak{z}}, a_b^{\mathfrak{z}}, s_{b+1}^{\mathfrak{z}}, r_b^{\mathfrak{z}})\}_{b=1}^{|B|}$ from $\mathcal{B}^{\mathfrak{z}}$ for $\mathfrak{z} \in \{\text{it}, \text{re}, \mathfrak{g}\}$
16:      Update $\pi^{\text{it}}$ with $B^{\text{it}}$, update $\pi^{\text{re}}$ with $B^{\text{re}}$, update $\mathfrak{g}$ with $B^{\mathfrak{g}}$.
17: **end for**

---

**Theorem 1** *SEREN converges to a stable solution in the agents' value functions.*

Theorem 1 is established by proving a series of results. Firstly, for a given $\pi^{\text{it}}$, we prove that the training process of Switcher converges (to its optimal value function). Secondly, we show that the system of the two learners Exploiter and Switcher jointly converges to their optimal value functions.

Our first result proves that the optimal value function of Switcher can be obtained as the limit point of a sequence of Bellman operations. We subsequently prove that the convergence result extends to linear function approximation. To begin, we firstly define the *optimal linear-projection*, $\Pi$, as:

$$\Pi \Lambda := \Phi r^*, \text{ where } r^* \in \arg\min_{r \in \mathbb{R}^p} \|\Phi r - \Lambda\| \tag{6}$$

for any function $\Lambda$, where $\Phi$ is a given matrix of $p$-dimensional linear features and $r$ is the corresponding weight vector.

Given a function $Q : \mathcal{S} \times \mathcal{A} \to \mathbb{R}$, $\forall \pi^{\text{re}}, \pi \in \Psi$ and $\forall \mathfrak{g}$, $\forall s_{\tau_k} \in \mathcal{S}$, we define the intervention operator $\mathcal{M}$ by $\mathcal{M}^{\pi^{\text{re}}}[Q^{\pi', \mathfrak{g}}(s_{\tau_k}, a)] := \mathbb{E}_{a_{\tau_k} \sim \pi^{\text{re}}(\cdot|s_{\tau_k})}[\mathcal{R}(s_{\tau_k}, a_{\tau_k}) - \beta + \gamma \sum_{s' \in \mathcal{S}} \mathcal{P}(s'; a_{\tau_k}, s) Q^{\pi', \mathfrak{g}}(s', a_{\tau_k})]$.[3] We denote by $\mathcal{M}Q^{\pi, \mathfrak{g}}$ the intervention operator acting on $Q^{\pi, \mathfrak{g}}$ when the immediate action is chosen according to an epsilon-greedy policy. Note that $\mathcal{M}$ applies similarly to the value functions $V : \mathcal{S} \to \mathbb{R}$.

**Proposition 1** *For a given Exploiter policy $\pi^{\text{it}} \in \Psi$, the training process of Switcher converges (i.e., $Q_{\mathfrak{g}} \to Q_{\mathfrak{g}}^{\star}$, see further details in Prop. 3). Moreover, given a set of linearly independent basis functions $\Phi = \{\phi_1, \ldots, \phi_p\}$ where $\phi_{1 \le k \le p} \in L_2$, the value function of the Switcher converges to a limit point, $r^{\star} \in \mathbb{R}^p$. $r^{\star}$ is the unique solution to $\Pi \mathcal{B}(\Phi r^{\star}) = \Phi r^{\star}$, where $\mathcal{B}$ is defined by: $\mathcal{B} \Lambda := \mathcal{R} + \gamma P \max\{\mathcal{M}\Lambda, \Lambda\}$. Moreover, $r^{\star}$ satisfies: $\|\Phi r^{\star} - Q^{\mathfrak{g}, \star}\| \le (1 - \gamma_{\mathfrak{g}}^2)^{-1/2} \|\Pi Q^{\mathfrak{g}, \star} - Q^{\mathfrak{g}, \star}\|$, where for any function $J : \mathcal{S} \to \mathbb{R}$ the functional $P$ is defined through $PJ := \sum_{s' \in \mathcal{S}} J(s') \mathcal{P}(s, a, s')$.*

Prop. 1 establishes the convergence guarantee of the training of Switcher (and additionally under the context of linear function approximation). The second statement bounds the proximity of the convergence point by the smallest approximation error that can be achieved given the choice of basis functions.

Having constructed a procedure to find the optimal Switcher policy, our next result characterises the Switcher policy $\mathfrak{g}$ and the times that Switcher must perform an intervention.

---

[3] Note for the term $\mathcal{M}Q$, the action input of the $Q$ function is a dummy variable decided by the operator $\mathcal{M}$.

**Proposition 2** *i) The Switcher intervention times are given by the following:*

$$\tau_k = \inf \left\{ \tau > \tau_{k-1} | \mathcal{M} v^{\mathfrak{g}} = v^{\mathfrak{g}} \right\} \tag{7}$$

*ii) Denote by $\mu_l(\mathfrak{g})$ the number of switch activations performed by the Switcher when $\max_{(s,a) \in \mathcal{S} \times \mathcal{A}} L(s,a) = l$ under $\mathfrak{g}$, then $\lim_{l \to 0} \mu_l(\mathfrak{g}) = 0$.*

Part i) of Prop. 2 characterises the distribution $\mathfrak{g}$. Moreover, given the corresponding value function, $v^{\mathfrak{g}}$, the switching times $\{\tau_k\}$ can be determined by evaluating if $\mathcal{M} v^{\mathfrak{g}} = v^{\mathfrak{g}}$ holds. Part ii) of Prop. 2 establishes that the number of switches performed by Switcher tends to 0 as the system uncertainty is reduced through the systematic exploration induced by Switcher, which consequently induces a natural exploration schedule based on the current system uncertainty.

## 5 Related Work

**Exploration-Exploitation Tradeoff** is a fundamental question in RL research, i.e. trading off finding higher reward states and exploiting known rewards. Existing approaches include directly injecting pure noise or certain parametric stochasticity into action choices during learning (Sutton & Barto, 2018; Lillicrap et al., 2015); using stochastic controllers regularised by the maximum entropy principle (Haarnoja et al., 2018); augmenting task rewards with synthetic exploration bonus / intrinsic reward (Stadie et al., 2015; Pathak et al., 2017; Sekar et al., 2020; Yu et al., 2024). Despite the simplicity, no existing methods explicitly learn an exploration policy for performing targeted exploration that exclusively maximise the expected uncertainty over the future trajectory. Moreover, most existing methods utilise one policy for capturing both the task-dependent optimal behaviour and the exploratory behaviour for efficient covering of the state space, yielding suboptimal learning in both aspects, whereas SEREN is able to disentangle the learning of the two policies with independently trained RL agents, leading to improved training for both the optimal policy and the exploration policy. Moreover, in contrast to *Reward free exploration* (also known as the task- or reward-agnostic exploration) (Zhang et al., 2020; Jin et al., 2020; Sekar et al., 2020), where the agent goes through independent exploration and RL stages (with respect to the extrinsic reward), SEREN performs exploration and learning simultaneously, eliminating the additional complexity involved with a pre-training (exploration) phase.

One prominent prior work on disentangling exploitation and exploration objectives is MULEX (Beyer et al., 2019), where separate value functions are learned for the environmental rewards and intrinsic exploratory rewards independently (both dependent with respect to environmental dynamics and behavioural policy). Similar to MULEX, SEREN is partially based on the motivation that learning separate value functions for environmental and intrinsic rewards would reduce interference of value learning, and also exclude the non-stationarity in the exploitation value function (based solely on environmental rewards). However, a key novelty of SEREN is the introduction of the impulse control mechanism for switching between exploitative and exploratory behaviour during the course of learning (Øksendal & Sulem, 2007; Mguni et al., 2022). Hence the SEREN agent is able to learn the optimal behaviour specified by the environmental reward function without human intervention that selects the suitable value function to act greedily with respect to.

**Uncertainty quantification in exploration** is an active field of research in RL. It is common practice to use the disagreement of the predictions over an ensemble of neural networks as the epistemic uncertainty to guide exploration Osband et al. (2016); Janner et al. (2019); Sekar et al. (2020); Lee et al. (2021). Connections between ensemble disagreement and information theory have been drawn showing that choosing actions that maximises the expected ensemble disagreement would maximally increase the information gain, improving the efficiency of exploration Sekar et al. (2020); O'Donoghue (2021). Other popular alternatives involve the prediction error of a discriminative dynamics model Schmidhuber et al. (1997); Pathak et al. (2017) and the predictive uncertainty given a generative dynamics model Ratzlaff et al. (2020); Jiang & Lu (2020). Future directions involve investigating instantiations of SEREN under different exploration bonuses.

**Relation to Markov games.** Our framework involves a system of two agents each with their individual objectives. Settings of this kind are formalised by Markov games (MG), a framework for studying self-interested agents that simultaneously act over time (Littman, 1994). In the standard MG setup, the actions

of *both* agents influence both each agent's rewards and the system dynamics. Therefore, each agent $i \in \{1, 2\}$ has its own reward function $\mathcal{R}_i : \mathcal{S} \times (\times_{i=1}^2 \mathcal{A}_i) \to \mathbb{R}$ and action set $\mathcal{A}_i$ and its goal is to maximise its *own* expected returns. The system dynamics, now influenced by both agents, are described by a transition probability $\mathcal{P} : \mathcal{S} \times (\times_{i=1}^2 \mathcal{A}_i) \times \mathcal{S} \to [0, 1]$. Unlike classical MGs, in our MG, Switcher does not intervene at each state but is allowed to assume control of the system at certain states which it decides using impulse controls. Our setup is related to stochastic differential games with impulse control (Mguni, 2018). However, our Markov Game differs markedly since it is nonzero-sum in which only one agent *assumes control* and is a discrete-time treatment.

**Policy Switching** Policy switching with respect to different reward function is an active field of research in reinforcement learning community. Its core concept could date back to the original *options* framework that partition the behavioural sequence in an online fashion for learning option-dependent optimal policy (Sutton et al., 1999). Some recent works focus on explicit construction of switching mechanism, similar to SEREN (Zhou et al., 2022; Jacq et al., 2022). In particular, Jacq et al. (2022) proposed Lazy-MDPs, which switch to random exploration policy during non-critical states for encouraging exploration. From an algorithmic perspective, Lazy-MDP can be interpreted as a degenerate form of SEREN such that the switching mechanism is manually constructed and the exploration policy is purely random. From a theoretical perspective, despite the switching mechanisms in both SEREN and Lazy-MDP are based on cost-minimisation, SEREN is developed based on the interplay of agents maximising different objective under a Markov game setting.

## 6 Experiments

We performed a series of experiments to demonstrate that SEREN's multi-player framework is able to improve the tradeoff between exploration and exploitation leading to marked improvement of the underling RL methods (all experimental details can be found in Appendix D).

Specifically, we wish to address the question of whether SEREN learns to improve performance of an underlying base RL learner by more efficiently locating higher reward states in MDPs with a) discrete b) continuous action spaces with different base learners (e.g., value-based and actor-critic). Moreover, we wish to empirically investigate if the non-stationary exploration reward structure negatively impacts the overall learning (hence justifying our choice of lower discounting value for the training of $\pi^{\mathrm{re}}$).

### 6.1 MiniGrid Experiments

We firstly demonstrate SEREN in combination with a standard DQN (Mnih et al., 2013). It is well known that DQN usually performs poorly in sparse-reward settings (Osband et al., 2016; Pathak et al., 2017). To this end, we choose the MiniGrid environments (Chevalier-Boisvert et al., 2018), where all transitions to non-goal states leads to zero reward. As we observe in Figure 2b, SEREN-DQN quickly learns to consistently navigate towards the goal state within 100 training episodes, whereas the standard DQN with $\epsilon$-greedy has failed to acquire a sensible policy over the 150 episodes. Hence we conclude that SEREN can be readily plugged into DQNs to deal with sparse-reward and/or goal-directed tasks (albeit not being our primary focus in the current paper).

### 6.2 MuJoCo Experiments

We now evaluate SEREN based on a stochastic policy algorithm, the Soft Actor-Critic (SAC Haarnoja et al. (2018)). SAC is an off-policy policy gradient algorithm, which augments the standard maximising return objective with an entropy maximisation term (Haarnoja et al., 2017), leading to an implicitly defined exploratory component. Due to the exploratory component and the off-policy nature, SAC is arguably one of the most sample-efficient algorithms, which makes any exploration improvements rather challenging. We evaluate our model, SEREN-SAC, on continuous control benchmarks from the MuJoCo suite (Figure 3a; Todorov et al. (2012)) to show that SEREN-SAC yields a more sample-efficient exploratory strategy than the implicit exploratory behaviour given by the maximum entropy regularisation in standard SAC. We further aim to disentangle the contribution of the improved performance by comparing the full SEREN-SAC with

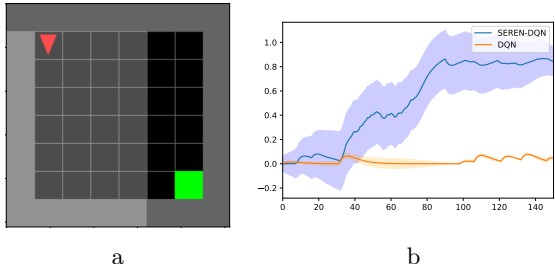

a b

Figure 2: DQN and SEREN-DQN in the MiniGrid-World Chevalier-Boisvert et al. (2018). (a) Graphical illustration of the "8x8" minigrid-world environment, only transitions into goal states (green block) lead to non-zero rewards; (b) SEREN-DQN quickly learns the optimal policy to the goal state while the standard DQN has not learned good policy over 150 episodes of training.

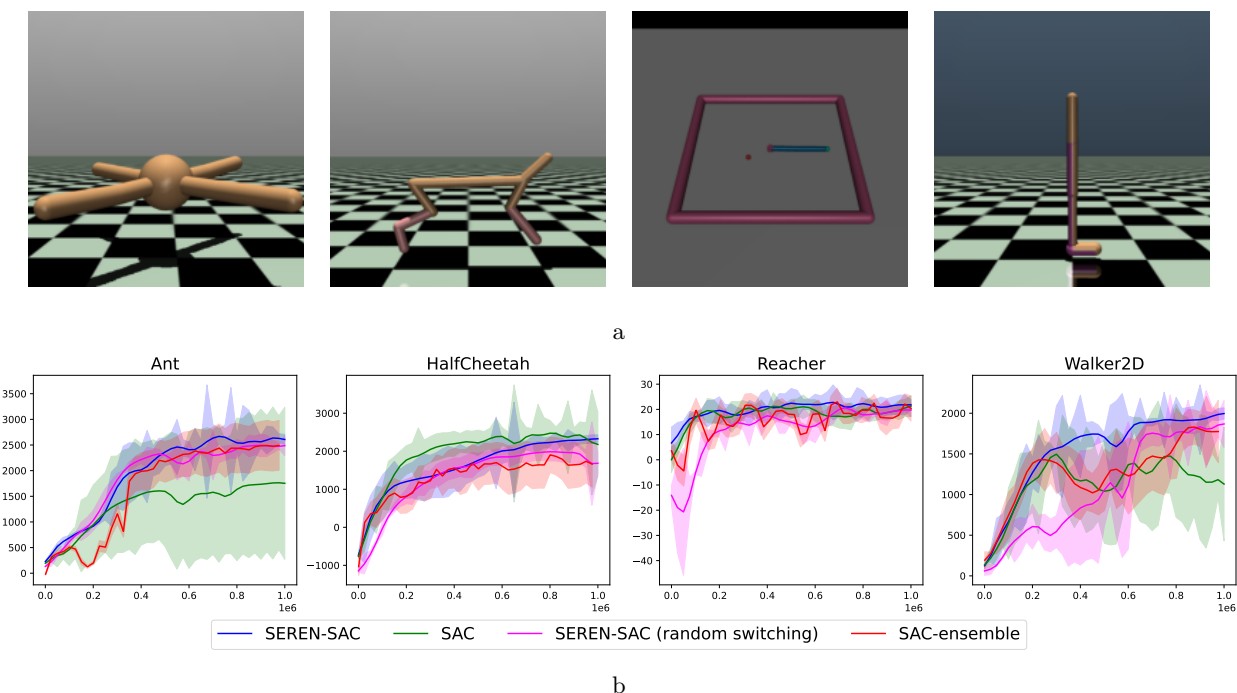

Figure 3: Evaluation of SEREN with the baseline SAC algorithms on the MuJoCo tasks (Todorov et al., 2012). Average evaluation returns (over 5 random seeds) of SEREN-SAC, SAC, and SAC-ensemble (see text) over $1 \times 10^6$ training steps.

associated degenerative alternatives: SAC, SEREN-SAC with random switching, and SAC-ensemble, where SAC-ensemble augments standard SAC with an ensemble-based value network, and utilises the ensemble epistemic uncertainty for computing the intrinsic reward for guiding switching (Equation 4).

From Figure 3, we observe that in 3 out of the 4 selected tasks, SEREN-SAC outperforms the baseline SAC agent in terms of both sample efficiency and the asymptotic performance, over the $10^6$ training steps. SEREN-SAC performs slightly worse than the baseline SAC on the HalfCheetah task in terms of sample efficiency of learning, but reaches similar asymptotic performance. Moreover, we note that across all 4 tasks, SEREN-SAC leads to more robust training, as indicated by the standard deviation of the evaluation scores over 5 random seeds throughout training. The gain may be attributed to the effective exploration by the Switcher, especially during the early phase of training, which facilitates the diversity of the off-policy replay buffer, hence enabling the identification of better solutions.

Finally, we evaluate the importance of the switching control mechanism for guiding exploration by comparing SEREN-SAC to SAC-intrinsic. In Figure 3 we observe that merely including the additive uncertainty-based intrinsic reward leads to worse performance across all tasks, and SEREN-SAC outperforms SAC-intrinsic on all 4 selected mujoco tasks. Hence, we empirically justify that the impulse switching control arbitration enables the learning of more targeted exploration policies comparing to the naive additive combination of the extrinsic reward and the uncertainty-based exploration bonus.

In order to further illustrate the importance of the impulse control switching mechanism, we implement an alternative version of SEREN-SAC, this time replacing the learned switching policy, $\mathfrak{g}$, with a random switching policy with manually defined decreasing switching probabilities. From Figure 3 we again observe that the full SEREN-SAC outperforms the alternative with random switching in all 4 tasks, demonstrating the effectiveness of the learned impulse control switching mechanism.

### 6.2.1 Ablation Studies on $\gamma_{\mathrm{re}}$

As discussed in Sec. 3, the discounting factor for the $\pi^{\mathrm{re}}$, $\gamma_{\mathrm{re}}$, needs to be set small to mitigate the negative effects of the non-stationarity reward structure in the training of $\pi^{\mathrm{re}}$. However, naively setting the explorative discounting factor too small would not yield good performance either, since the resulting $\pi^{\mathrm{re}}$ would choose actions primarily dependent on the immediate epistemic uncertainty, hence failing to generate targeted exploratory behaviour towards areas of high uncertainties. Here we empirically justify our hypothesis by performing an ablation study on the effect of the value of the discounting factor for the training of $\pi^{\mathrm{re}}$. By examining the asymptotic performance given $1 \times 10^6$ training steps (Table 1), we see that setting the discounting factor too large or too small both induce worse performance, whereas an intermediate values of $\gamma_{\mathrm{re}}$ (0.6) yields the best performance in terms of averaged evaluation return. Noticeably, we also observe that setting $\gamma_{\mathrm{re}}$ too large or too small makes the training less robust with respect to the random seed, leading to increased variability in the evaluation performance.

Table 1: Ablation studies on the effects of the discounting factor values of $\pi^{\mathrm{re}}$ ($\gamma_{\mathrm{re}}$).

|  | Ant | HalfCheetah | Reacher | Walker2D |
|---|---|---|---|---|
| SEREN-SAC ($\gamma_{\mathrm{re}} = 0.6$) | **2607.6 ± 99.0** | **2327.8 ± 102.5** | **21.8 ± 2.0** | **1996.1 ± 69.9** |
| SEREN-SAC ($\gamma_{\mathrm{re}} = 0.1$) | 1853.7 ± 1362.7 | 2205.4 ± 846.2 | 20.2 ± 4.0 | 1169.7 ± 762.9 |
| SEREN-SAC ($\gamma_{\mathrm{re}} = 0.98$) | 2477.9 ± 128.9 | 1944.9 ± 587.0 | 18.6 ± 0.8 | 1720.7 ± 177.6 |

### 6.2.2 Asymptotic Bias Towards Full Exploitation

We prove in Proposition 2 that as the agent's quantification of uncertainty over the state space (indicated by the $\mathbb{E}_{(s,a) \in \mathcal{S} \times \mathcal{A}}[L(s,a)]$, equation 4) decreases given more exploration, the number of switching activations performed by Switcher converges to 0 asymptotically, hence the overall behaviour gradually shifts towards pure exploitation given more training. We observe this behaviour empirically with SEREN-SAC on the mujoco tasks (Figure 4).

### 6.3 Atari Experiments

We further evaluate SEREN on a set of sample-constrained discrete control tasks, the Atari 100K benchmark (Kaiser et al., 2019). Here we instantiate SEREN based on DQN (Mnih et al., 2013), with additional components, including double Q-learning (Van Hasselt et al., 2016), dueling network for value estimation (Wang et al., 2016), and multi-step TD-target (Mnih et al., 2016). The resulting DQN instantiated with these additional features are referred to as the Efficient-DQN (Kostrikov et al., 2020). We additionally utilise the image augmentation techniques for training the DQN proposed in DrQ (Kostrikov et al., 2020) (we only use the "intensity" augmentation instead of all augmentation types as in Kostrikov et al. (2020)). We apply the resulting model, SEREN-Eff-DQN on all games in the Atari 100K benchmark, and we evaluate the performance given 100K training steps. We follow the evaluation procedures outlined in Kaiser et al. (2019). In Figure 5, we show that SEREN-Eff-DQN outperforms all selected baselines (see Appendix D) in terms of

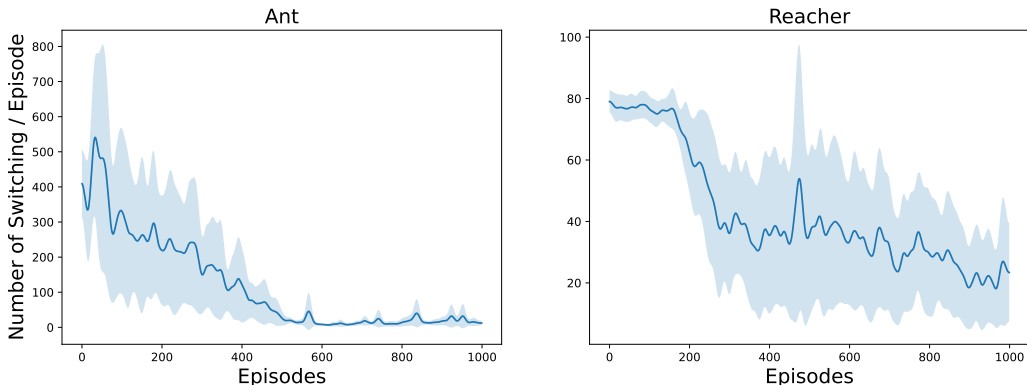

Figure 4: Asymptotic decrease of switching actions given more training.

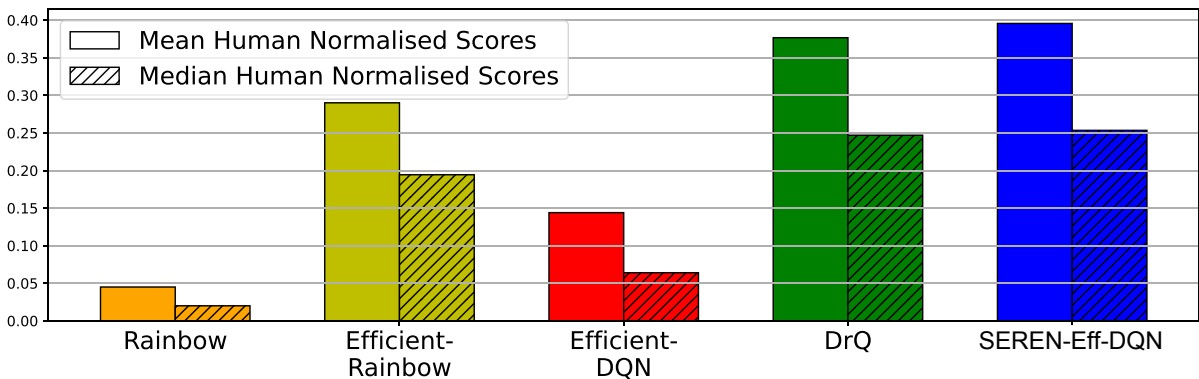

Figure 5: Mean (normal bars) and median (shaded bars) human normalised scores of evaluations of SEREN-Eff-DQN and selected baselines (see Appendix D) on Atari 100K benchmarks.

both the median and mean human normalized returns for all SEREN-Eff-DQN and selected baselines, hence again indicating the improvement brought by the SEREN framework. While SEREN-Eff-DQN is performing fairly similarly to DrQ, the plug & play nature of our approach means that using other base algorithms can further improve the performance without significant redesign. Full experimental setup and results (for all games) can be found in Appendix D and E.

## 7 Discussion

We introduced SEREN, a plug-and-play framework that seeks to learn the optimal arbitration between exploitative and exploratory behaviours using an impulse control mechanism. SEREN can be readily combined with existing value-based and actor-critic algorithms, here we demonstrate the instantiations of SEREN given DQN and SAC, but more combinations can be considered for future works. We formulate the problem of the arbitration between the exploration and the exploitation policies under a Markov game framework, where Exploiter seeks to only maximise the cumulative return and Switcher, consisting of an exploration policy, $\pi^{\mathrm{re}}$, and an impulse control switching policy, $\mathfrak{g}$, seeks to learn the optimal exploration timing and exploration behaviour, in order to minimise the epistemic uncertainty of the value estimates of Exploiter (reflecting the level of information about the environment acquired by the joint system) over the state space. We provide theoretical justification for the convergence of SEREN to the optimal achievable value estimates with linear function approximation. We demonstrate the utility of SEREN through extensive experimental studies on continuous control benchmarks. When implemented with state-of-the-art policy gradient algorithms (SAC),

we show that the SEREN-augmented agents consistently yield improvement in terms of sample efficiency and asymptotic performance with respect to the baseline agents. We also showed that SEREN can be combined with value-based algorithms such as DQN, to yield improvement on the Atari 100K benchmarks over competitive baseline algorithms.

Behaviourally, animals tend to sacrifice short-term rewards to obtain information gain in uncertain environments (Bromberg-Martin & Hikosaka, 2009; Gottlieb et al., 2013). Blanchard et al. (2015) demonstrated that OFC neurons have firing correlated with both information value and primary value signals. Instead of integrating these variables to code subjective value, they found that OFC neurons tend to encode the two signals in an orthogonal manner. Hence providing stronger biological plausibility, supporting the independent representation and learning paradigm in SEREN over the joint encoding scheme in standard intrinsic reward models Pathak et al. (2017).

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

## Appendix for Systematic Exploration and Exploitation via a Markov Game with Impulse Control

## Appendix Table of Contents

## A    Notation & Assumptions

We assume that $\mathcal{S}$ is defined on a probability space $(\Omega, \mathcal{F}, \mathbb{P})$ and any $s \in \mathcal{S}$ is measurable with respect to the Borel $\sigma$-algebra associated with $\mathbb{R}^p$. We denote the $\sigma$-algebra of events generated by $\{s_t\}_{t \geq 0}$ by $\mathcal{F}_t \subset \mathcal{F}$. In what follows, we denote by $(\mathcal{V}, \|\|)$ any finite normed vector space and by $\mathcal{H}$ the set of all measurable functions.

For notational simplicity, we denote the value/action functions for Exploiter and Explorer as $v_1/Q_1$ and $v_2/Q_2$, respectively.

The results of the paper are built under the following assumptions which are standard within RL and stochastic approximation methods:

**Assumption 1** The stochastic process governing the system dynamics is ergodic, that is the process is stationary and every invariant random variable of $\{s_t\}_{t \geq 0}$ is equal to a constant with probability 1.

**Assumption 2** The constituent functions of the players' objectives $R$ and $L$ are in $L_2$ (square-integrable functions).

**Assumption 3** For any positive scalar $c$, there exists a scalar $\mu_c$ such that for all $s \in \mathcal{S}$ and for any $t \in \mathbb{N}$ we have: $\mathbb{E}\left[1 + \|s_t\|^c | s_0 = s\right] \leq \mu_c(1 + \|s\|^c)$.

**Assumption 4** There exists scalars $C_1$ and $c_1$ such that for any function $J$ satisfying $|J(s)| \leq C_2(1 + \|s\|^{c_2})$ for some scalars $c_2$ and $C_2$ we have that: $\sum_{t=0}^{\infty} |\mathbb{E}\left[J(s_t)|s_0 = s\right] - \mathbb{E}[J(s_0)]| \leq C_1 C_2 (1 + \|s_t\|^{c_1 c_2})$.

**Assumption 5** There exists scalars $c$ and $C$ such that for any $s \in \mathcal{S}$ we have that: $|J(z, \cdot)| \leq C(1 + \|z\|^c)$ for $J \in \{R, L\}$.

We also make the following finiteness assumption on set of switching control policies for Switcher:

**Assumption 6** For any policy $\mathfrak{g}_c$, the total number of interventions is given by $K < \infty$.

In what follows, we denote by $\boldsymbol{\mathcal{A}} := \mathcal{A}^{\mathrm{it}} \times \mathcal{A}^{\mathrm{re}}$ and by $\boldsymbol{\Psi} := \Psi^{\mathrm{it}} \times \Psi^{\mathrm{re}} \times G$ where $G$ is the policy set of the Switcher's policy $\mathfrak{g}$.

# B    Proof of Technical Results

We begin the analysis with some preliminary lemmata and definitions which are useful for proving the main results.

**Definition 1** *An operator $T : \mathcal{V} \to \mathcal{V}$ is said to be a **contraction** w.r.t a norm $\|\cdot\|$ if there exists a constant $c \in [0, 1[$ such that for any $V_1, V_2 \in \mathcal{V}$ we have that:*

$$\|TV_1 - TV_2\| \leq c\|V_1 - V_2\|. \tag{8}$$

**Definition 2** *An operator $T : \mathcal{V} \to \mathcal{V}$ is **non-expansive** if $\forall V_1, V_2 \in \mathcal{V}$ we have:*

$$\|TV_1 - TV_2\| \leq \|V_1 - V_2\|. \tag{9}$$

**Lemma 1** *For any $f : \mathcal{V} \to \mathbb{R} : \mathcal{V} \to \mathbb{R}$, we have that:*

$$\left\| \max_{a \in \mathcal{V}} f(a) - \max_{a \in \mathcal{V}} g(a) \right\| \leq \max_{a \in \mathcal{V}} \|f(a) - g(a)\|. \tag{10}$$

**Proof 1** *We restate the proof given in Mguni (2019):*

$$f(a) - g(a) \leq \|f(a) - g(a)\| \tag{11}$$

$$\implies f(a) \leq \|f(a) - g(a)\| + g(a) \tag{12}$$

$$\implies \max_{a \in \mathcal{V}} f(a) \leq \max_{a \in \mathcal{V}}\{\|f(a) - g(a)\| + g(a)\} \leq \max_{a \in \mathcal{V}} \|f(a) - g(a)\| + \max_{a \in \mathcal{V}} g(a). \tag{13}$$

*Deducting $\max_{a \in \mathcal{V}} g(a)$ from both sides of (13) yields:*

$$\max_{a \in \mathcal{V}} f(a) - \max_{a \in \mathcal{V}} g(a) \leq \max_{a \in \mathcal{V}} \|f(a) - g(a)\|. \tag{14}$$

*After reversing the roles of $f$ and $g$ and redoing steps (12) - (13), we deduce the desired result since the RHS of (14) is unchanged.*

**Lemma 2** *The probability transition kernel $P$ is non-expansive, that is:*

$$\|PV_1 - PV_2\| \leq \|V_1 - V_2\|. \tag{15}$$

**Proof 2** *The result is well-known e.g. Tsitsiklis & Van Roy (1999). We give a proof using the Tonelli-Fubini theorem and the iterated law of expectations, we have that:*

$$\|PJ\|^2 = \mathbb{E}\left[(PJ)^2[s_0]\right] = \mathbb{E}\left[[\mathbb{E}[J[s_1]|s_0]]^2\right] \leq \mathbb{E}\left[\mathbb{E}\left[J^2[s_1]|s_0\right]\right] = \mathbb{E}\left[J^2[s_1]\right] = \|J\|^2,$$

*where we have used Jensen's inequality to generate the inequality. This completes the proof.*

## Proof of Theorem 1

We begin by proving the following result:

**Proposition 3** *Define by $\mathcal{R}^{\mathrm{re}}(s_t, a_t^{\mathrm{it}}, a_t^{\mathrm{re}}, g_t) := R^{\mathrm{re}}\left(s_t, a_t^{\mathrm{it}}, a_t^{\mathrm{re}}\right) - \beta \cdot g_t$ and consider the following Q learning variant:*

$$Q_{2,t+1}(s_t, a_t^{\mathrm{it}}, a_t^{\mathrm{re}}) = Q_{2,t}(s_t, a_t^{\mathrm{it}}, a_t^{\mathrm{re}})$$

$$+ \alpha_t(s_t, a_t^{\mathrm{re}})\left[\max\left\{\mathcal{M}^{\pi,\mathfrak{g}}Q_{2,t}(s_t, a_t^{\mathrm{it}}, a_t^{\mathrm{re}}), \mathcal{R}^{\mathrm{re}}(s_t, a_t^{\mathrm{it}}, a_t^{\mathrm{re}}) + \gamma\max_{a' \in \mathcal{A}} Q_{2,t}(s_{t+1}, a_t^{\mathrm{it}}, a')\right\} - Q_{2,t}(s_t, a_t^{\mathrm{it}}, a_t^{\mathrm{re}})\right],$$

*then for a fixed **Exploiter** policy $\pi^{\mathrm{it}}$ and for a fixed $L$, $Q_{2,t}(s)$ converges to $Q_2^*$ with probability 1, where $s_t, s_{t+1} \in \mathcal{S}$ and $a_t^{\mathrm{it}} \sim \pi^{\mathrm{it}}(\cdot|s_t)$ is **Exploiter**'s action.*

**Proof 3** *Recall that for a function $Q : \mathcal{S} \times \mathcal{A} \to \mathbb{R}$, $\forall \pi^{\mathrm{re}}, \pi \in \Psi$ and $\forall \mathfrak{g}$, $\forall s_{\tau_k} \in \mathcal{S}$, we define the intervention operator $\mathcal{M}$ by $\mathcal{M}^{\pi^{\mathrm{re}}}[Q^{\pi', \mathfrak{g}}(s_{\tau_k}, a)] := \mathbb{E}_{a_{\tau_k} \sim \pi^{\mathrm{re}}(\cdot|s_{\tau_k})}[\mathcal{R}(s_{\tau_k}, a_{\tau_k}) - \beta + \gamma \sum_{s' \in \mathcal{S}} \mathcal{P}(s'; a_{\tau_k}, s) Q^{\pi', \mathfrak{g}}(s', a_{\tau_k})]$. We denote by $\mathcal{M}Q^{\pi, \mathfrak{g}}$ the intervention operator acting on $Q^{\pi, \mathfrak{g}}$ when the immediate action is chosen according to an epsilon-greedy policy. Note that $\mathcal{M}$ applies similarly to the value functions by defining the Bellman operator $T$ of acting on the value function $v_2^{\pi^{\mathrm{it}}, \pi^{\mathrm{re}}} : \mathcal{S} \to \mathbb{R}$ by*

$$Tv(s_{\tau_k}) := \max \left\{ \mathcal{M}v(s_{\tau_k}), \left[ \mathcal{R}^{\mathrm{re}}(s_{\tau_k}, \boldsymbol{a}) + \gamma \max_{\boldsymbol{a} \in \mathcal{A}} \sum_{s' \in \mathcal{S}} \mathcal{P}(s'; \boldsymbol{a}, s_{\tau_k}) v(s') \right] \right\} \tag{16}$$

*Our first result proves that the operator $T$ is a contraction operator. First let us recall that the switching time $\tau_k$ is defined recursively $\tau_k = \inf\{t > \tau_{k-1} | s_t \in A, \tau_k \in \mathcal{F}_t\}$ where $A = \{s \in \mathcal{S} | \mathfrak{g}(s_t) = 1\}$. To this end, we show that the following bounds holds:*

**Lemma 3** *The Bellman operator $T$ is a contraction, that is the following bound holds:*

$$\|T\psi - T\psi'\| \le \gamma \|\psi - \psi'\|.$$

*In what follows and for the remainder of the script, we employ the following shorthands:*

$$\mathcal{P}_{ss'}^{\boldsymbol{a}} =: \sum_{s' \in \mathcal{S}} \mathcal{P}(s'; \boldsymbol{a}, s), \quad \mathcal{P}_{ss'}^{\boldsymbol{\pi}} =: \sum_{\boldsymbol{a} \in \mathcal{A}} \boldsymbol{\pi}(\boldsymbol{a}|s) \mathcal{P}_{ss'}^{\boldsymbol{a}}$$

*To prove that $T$ is a contraction, we consider the three cases produced by equation 16, that is to say we prove the following statements:*

*i)*
$$\left| \mathcal{R}^{\mathrm{re}}(s_t, \boldsymbol{a}) + \gamma \max_{\boldsymbol{a} \in \mathcal{A}} \mathcal{P}_{s's_t}^{\boldsymbol{a}} v(s') - \left( \mathcal{R}^{\mathrm{re}}(s_t, \boldsymbol{a}) + \gamma \max_{\boldsymbol{a} \in \mathcal{A}} \mathcal{P}_{s's_t}^{\boldsymbol{a}} v'(s') \right) \right| \le \gamma \|v - v'\|$$

*ii)*
$$\|\mathcal{M}v - \mathcal{M}v'\| \le \gamma \|v - v'\|, \qquad \text{(and hence } \mathcal{M} \text{ is a contraction).}$$

*iii)*
$$\left\| \mathcal{M}v - \left[ \mathcal{R}^{\mathrm{re}}(\cdot, \boldsymbol{a}) + \gamma \max_{\boldsymbol{a} \in \mathcal{A}} \mathcal{P}^{\boldsymbol{a}} v' \right] \right\| \le \gamma \|v - v'\|.$$

*We begin by proving i).*

*Indeed, for any $a^{\mathrm{it}} \in \mathcal{A}^{\mathrm{it}}$ and $\forall s_t \in \mathcal{S}, \forall s' \in \mathcal{S}$ we have that*

$$\left| \mathcal{R}^{\mathrm{re}}(s_{\tau_k}, \boldsymbol{a}) + \gamma \mathcal{P}_{s's_t}^{\boldsymbol{\pi}} v(s') - \left[ \mathcal{R}^{\mathrm{re}}(s_{\tau_k}, \boldsymbol{a}) + \gamma \max_{\boldsymbol{a} \in \mathcal{A}} \mathcal{P}_{s's_t}^{\boldsymbol{a}} v'(s') \right] \right|$$

$$\le \max_{\boldsymbol{a} \in \mathcal{A}} \left| \gamma \mathcal{P}_{s's_t}^{\boldsymbol{a}} v(s') - \gamma \mathcal{P}_{s's_t}^{\boldsymbol{a}} v'(s', \cdot) \right|$$

$$\le \gamma \|Pv - Pv'\|$$

$$\le \gamma \|v - v'\|,$$

*again using the fact that $P$ is non-expansive and Lemma 1.*

*We now prove ii).*

*For any $\tau \in \mathcal{F}$, define by $\tau' = \inf\{t > \tau | s_t \in A, \tau \in \mathcal{F}_t\}$. Now using the definition of $\mathcal{M}$ we have that for any $s_\tau \in \mathcal{S}$*

$$|(\mathcal{M}v - \mathcal{M}v')(s_\tau)|$$

$$\le \max_{\boldsymbol{a}_\tau \in \mathcal{A}} \left| \mathcal{R}^{\mathrm{re}}(s_\tau, \boldsymbol{a}_\tau) - \beta \mathbf{1}_{\mathcal{A}^{\mathrm{re}}}(a_t^{\mathrm{re}}) + \gamma \mathcal{P}_{s's_\tau}^{\boldsymbol{\pi}} \mathcal{P}^{\boldsymbol{a}} v(s_\tau) - \left( \mathcal{R}^{\mathrm{re}}(s_\tau, \boldsymbol{a}_\tau) - \beta \mathbf{1}_{\mathcal{A}^{\mathrm{re}}}(a_t^{\mathrm{re}}) + \gamma \mathcal{P}_{s's_\tau}^{\boldsymbol{\pi}} \mathcal{P}^{\boldsymbol{a}} v'(s_\tau) \right) \right|$$

$$= \gamma \left| \mathcal{P}_{s's_\tau}^{\boldsymbol{\pi}} \mathcal{P}^{\boldsymbol{a}} v(s_\tau) - \mathcal{P}_{s's_\tau}^{\boldsymbol{\pi}} \mathcal{P}^{\boldsymbol{a}} v'(s_\tau) \right|$$

$$\le \gamma \|Pv - Pv'\|$$

$$\le \gamma \|v - v'\|,$$

*using the fact that $P$ is non-expansive. The result can then be deduced easily by applying max on both sides.*

*We now prove iii). We split the proof of the statement into two cases:*

**Case 1:**

$$\mathcal{M}v(s_\tau) - \left( \mathcal{R}^{\mathrm{re}}(s_\tau, \boldsymbol{a}_\tau) + \gamma \max_{\boldsymbol{a} \in \boldsymbol{\mathcal{A}}} \mathcal{P}^{\boldsymbol{a}}_{s's_\tau} v'(s') \right) < 0. \tag{17}$$

*We now observe the following:*

$$\mathcal{M}v(s_\tau) - \mathcal{R}^{\mathrm{re}}(s_\tau, \boldsymbol{a}_\tau) + \gamma \max_{\boldsymbol{a} \in \boldsymbol{\mathcal{A}}} \mathcal{P}^{\boldsymbol{a}}_{s's_\tau} v'(s')$$

$$\leq \max \left\{ \mathcal{R}^{\mathrm{re}}(s_\tau, \boldsymbol{a}_\tau) + \gamma \mathcal{P}^{\boldsymbol{\pi}}_{s's_\tau} \mathcal{P}^{\boldsymbol{a}} v(s'), \mathcal{M}v(s_\tau) \right\} - \mathcal{R}^{\mathrm{re}}(s_\tau, \boldsymbol{a}_\tau) + \gamma \max_{\boldsymbol{a} \in \boldsymbol{\mathcal{A}}} \mathcal{P}^{\boldsymbol{a}}_{s's_\tau} v'(s')$$

$$\leq \left| \max \left\{ \mathcal{R}^{\mathrm{re}}(s_\tau, \boldsymbol{a}_\tau) + \gamma \mathcal{P}^{\boldsymbol{\pi}}_{s's_\tau} \mathcal{P}^{\boldsymbol{a}} v(s'), \mathcal{M}v(s_\tau) \right\} - \max \left\{ \mathcal{R}^{\mathrm{re}}(s_\tau, \boldsymbol{a}_\tau) + \gamma \max_{\boldsymbol{a} \in \boldsymbol{\mathcal{A}}} \mathcal{P}^{\boldsymbol{a}}_{s's_\tau} v'(s'), \mathcal{M}v(s_\tau) \right\} \right.$$

$$\left. + \max \left\{ \mathcal{R}^{\mathrm{re}}(s_\tau, \boldsymbol{a}_\tau) + \gamma \max_{\boldsymbol{a} \in \boldsymbol{\mathcal{A}}} \mathcal{P}^{\boldsymbol{a}}_{s's_\tau} v'(s'), \mathcal{M}v(s_\tau) \right\} - \mathcal{R}^{\mathrm{re}}(s_\tau, \boldsymbol{a}_\tau) + \gamma \max_{\boldsymbol{a} \in \boldsymbol{\mathcal{A}}} \mathcal{P}^{\boldsymbol{a}}_{s's_\tau} v'(s') \right|$$

$$\leq \left| \max \left\{ \mathcal{R}^{\mathrm{re}}(s_\tau, \boldsymbol{a}_\tau) + \gamma \max_{\boldsymbol{a} \in \boldsymbol{\mathcal{A}}} \mathcal{P}^{\boldsymbol{a}}_{s's_\tau} v(s'), \mathcal{M}v(s_\tau) \right\} - \max \left\{ \mathcal{R}^{\mathrm{re}}(s_\tau, \boldsymbol{a}_\tau) + \gamma \max_{\boldsymbol{a} \in \boldsymbol{\mathcal{A}}} \mathcal{P}^{\boldsymbol{a}}_{s's_\tau} v'(s'), \mathcal{M}v(s_\tau) \right\} \right|$$

$$+ \left| \max \left\{ \mathcal{R}^{\mathrm{re}}(s_\tau, \boldsymbol{a}_\tau) + \gamma \max_{\boldsymbol{a} \in \boldsymbol{\mathcal{A}}} \mathcal{P}^{\boldsymbol{a}}_{s's_\tau} v'(s'), \mathcal{M}v(s_\tau) \right\} - \mathcal{R}^{\mathrm{re}}(s_\tau, \boldsymbol{a}_\tau) + \gamma \max_{\boldsymbol{a} \in \boldsymbol{\mathcal{A}}} \mathcal{P}^{\boldsymbol{a}}_{s's_\tau} v'(s') \right|$$

$$\leq \gamma \max_{\boldsymbol{a} \in \boldsymbol{\mathcal{A}}} \left| \mathcal{P}^{\boldsymbol{\pi}}_{s's_\tau} \mathcal{P}^{\boldsymbol{a}} v(s') - \mathcal{P}^{\boldsymbol{\pi}}_{s's_\tau} \mathcal{P}^{\boldsymbol{a}} v'(s') \right|$$

$$+ \left| \max \left\{ 0, \mathcal{M}v(s_\tau) - \left( \mathcal{R}^{\mathrm{re}}(s_\tau, \boldsymbol{a}_\tau) + \gamma \max_{\boldsymbol{a} \in \boldsymbol{\mathcal{A}}} \mathcal{P}^{\boldsymbol{a}}_{s's_\tau} v'(s') \right) \right\} \right|$$

$$\leq \gamma \| Pv - Pv' \|$$

$$\leq \gamma \| v - v' \|,$$

*where we have used the fact that for any scalars $a, b, c$ we have that $|\max\{a, b\} - \max\{b, c\}| \leq |a - c|$ and the non-expansiveness of $P$.*

**Case 2:**

$$\mathcal{M}v(s_\tau) - \left( \mathcal{R}^{\mathrm{re}}(s_\tau, \boldsymbol{a}_\tau) + \gamma \max_{\boldsymbol{a} \in \boldsymbol{\mathcal{A}}} \mathcal{P}^{\boldsymbol{a}}_{s's_\tau} v'(s') \right) \geq 0.$$

*For this case, we have that*

$$\mathcal{M}v(s_\tau) - \left( \mathcal{R}^{\mathrm{re}}(s_\tau, \boldsymbol{a}_\tau) + \gamma \max_{\boldsymbol{a} \in \boldsymbol{\mathcal{A}}} \mathcal{P}^{\boldsymbol{a}}_{s's_\tau} v'(s') \right)$$

$$\leq \mathcal{M}v(s_\tau) - \left( \mathcal{R}^{\mathrm{re}}(s_\tau, \boldsymbol{a}_\tau) + \gamma \max_{\boldsymbol{a} \in \boldsymbol{\mathcal{A}}} \mathcal{P}^{\boldsymbol{a}}_{s's_\tau} v'(s') \right) + \beta \mathbf{1}_{\mathcal{A}^{\mathrm{re}}}(a^{\mathrm{re}}_t)$$

$$\leq \mathcal{R}^{\mathrm{re}}(s_\tau, \boldsymbol{a}_\tau) - \beta \mathbf{1}_{\mathcal{A}^{\mathrm{re}}}(a^{\mathrm{re}}_t) + \gamma \mathcal{P}^{\boldsymbol{\pi}}_{s's_\tau} \mathcal{P}^{\boldsymbol{a}} v(s')$$

$$- \left( \mathcal{R}^{\mathrm{re}}(s_\tau, \boldsymbol{a}_\tau) - \beta \mathbf{1}_{\mathcal{A}^{\mathrm{re}}}(a^{\mathrm{re}}_t) + \gamma \max_{\boldsymbol{a} \in \boldsymbol{\mathcal{A}}} \mathcal{P}^{\boldsymbol{a}}_{s's_\tau} v'(s') \right)$$

$$\leq \gamma \max_{\boldsymbol{a} \in \boldsymbol{\mathcal{A}}} \left| \mathcal{P}^{\boldsymbol{\pi}}_{s's_\tau} \mathcal{P}^{\boldsymbol{a}} \left( v(s') - v'(s') \right) \right|$$

$$\leq \gamma \left| v(s') - v'(s') \right|$$

$$\leq \gamma \| v - v' \|,$$

*again using the fact that $P$ is non-expansive. Hence we have succeeded in showing that for any $v, v' \in L_2$ we have that*

$$\left\| \mathcal{M}v - \max_{\boldsymbol{a} \in \mathcal{A}} \left[ \mathcal{R}(\cdot, a) + \gamma \mathcal{P}^{\boldsymbol{a}} v' \right] \right\| \leq \gamma \left\| v - v' \right\|. \tag{18}$$

*Gathering the results of the three cases gives the desired result.*

*We now make use of the following result:*

**Theorem 2 (Theorem 1, pg 4 in Jaakkola et al. (1994))** *Let $\Xi_t(s)$ be a random process that takes values in $\mathbb{R}^n$ and given by the following:*

$$\Xi_{2,t+1}(s) = (1 - \alpha_t(s)) \Xi_{2,t}(s) \alpha_t(s) L_t(s), \tag{19}$$

*then $\Xi_t(s)$ converges to 0 with probability 1 under the following conditions:*

i) $0 \leq \alpha_t \leq 1, \sum_t \alpha_t = \infty$ and $\sum_t \alpha_t < \infty$

ii) $\|\mathbb{E}[L_t|\mathcal{F}_t]\| \leq \gamma \|\Xi_t\|$, with $\gamma < 1$;

iii) $\mathrm{Var}\left[ L_t|\mathcal{F}_t \right] \leq c(1 + \|\Xi_t\|^2)$ for some $c > 0$.

**Proof 4** *To prove the convergence in Theorem 1, we show (i) - (iii) hold. Condition (i) holds by choice of learning rate. It therefore remains to prove (ii) - (iii). We first prove (ii). For this, we consider our variant of the Q-learning update rule:*

$$Q_{2,t+1}(s_t, \boldsymbol{a}_t) = Q_{2,t}(s_t, \boldsymbol{a}_t) + \alpha_t(s_t, \boldsymbol{a}_t) \left[ \max \left\{ \mathcal{M}^{\boldsymbol{\pi}} Q_{2,t}(s_{\tau_k}, \boldsymbol{a}), \mathcal{R}^{\mathrm{re}}(s_{\tau_k}, \boldsymbol{a}) + \gamma \max_{a' \in \mathcal{A}} Q_{2,t}(s', \boldsymbol{a}') \right\} - Q_{2,t}(s_t, \boldsymbol{a}_t) \right].$$

*After subtracting $Q_2^\star(s_t, \boldsymbol{a}_t)$ from both sides and some manipulation we obtain that:*

$$\begin{aligned} \Xi_{2,t+1}(s_t, \boldsymbol{a}_t) = {} & (1 - \alpha_t(s_t, \boldsymbol{a}_t)) \Xi_{2,t}(s_t, \boldsymbol{a}_t) \\ & + \alpha_t(s_t, \boldsymbol{a}_t) \left[ \max \left\{ \mathcal{M}^{\boldsymbol{\pi}} Q_{2,t}(s_{\tau_k}, \boldsymbol{a}), \mathcal{R}^{\mathrm{re}}(s_{\tau_k}, \boldsymbol{a}) + \gamma \max_{a' \in \mathcal{A}} Q_{2,t}(s', \boldsymbol{a}') \right\} - Q_2^\star(s_t, \boldsymbol{a}_t) \right], \end{aligned}$$

*where $\Xi_{2,t}(s_t, \boldsymbol{a}_t) := Q_{2,t}(s_t, \boldsymbol{a}_t) - Q_{2,t}^\star(s_t, \boldsymbol{a}_t)$.*

*Let us now define by*

$$\mathfrak{L}_t(s_{\tau_k}, \boldsymbol{a}) := \max \left\{ \mathcal{M}^{\boldsymbol{\pi}} Q_{2,t}(s_{\tau_k}, \boldsymbol{a}), \mathcal{R}^{\mathrm{re}}(s_{\tau_k}, \boldsymbol{a}) + \gamma \max_{a' \in \mathcal{A}} Q_{2,t}(s', \boldsymbol{a}') \right\} - Q_2^\star(s_t, a).$$

*Then*

$$\Xi_{2,t+1}(s_t, \boldsymbol{a}_t) = (1 - \alpha_t(s_t, \boldsymbol{a}_t)) \Xi_{2,t}(s_t, \boldsymbol{a}_t) + \alpha_t(s_t, \boldsymbol{a}_t) \left[ \mathfrak{L}_t(s_{\tau_k}, a) \right]. \tag{20}$$

*We now observe that*

$$\begin{aligned} \mathbb{E}\left[ \mathfrak{L}_t(s_{\tau_k}, \boldsymbol{a}) | \mathcal{F}_t \right] &= \sum_{s' \in \mathcal{S}} \mathcal{P}(s'; \boldsymbol{a}, s_{\tau_k}) \max \left\{ \mathcal{M}^{\boldsymbol{\pi}} Q_{2,t}(s_{\tau_k}, \boldsymbol{a}), \mathcal{R}^{\mathrm{re}}(s_{\tau_k}, \boldsymbol{a}) + \gamma \max_{a' \in \mathcal{A}} Q_{2,t}(s', \boldsymbol{a}') \right\} - Q_2^\star(s_{\tau_k}, \boldsymbol{a}) \\ &= TQ_{2,t}(s, \boldsymbol{a}) - Q_2^\star(s, \boldsymbol{a}). \end{aligned} \tag{21}$$

*Now, using the fixed point property that implies $Q_2^\star = TQ_2^\star$, we find that*

$$\begin{aligned} \mathbb{E}\left[ \mathfrak{L}_t(s_{\tau_k}, \boldsymbol{a}) | \mathcal{F}_t \right] &= TQ_{2,t}(s, \boldsymbol{a}) - TQ_2^\star(s, \boldsymbol{a}) \\ &\leq \| TQ_{2,t} - TQ_2^\star \| \\ &\leq \gamma \| Q_{2,t} - Q_2^\star \|_\infty = \gamma \| \Xi_t \|_\infty. \end{aligned} \tag{22}$$

*using the contraction property of $T$ established in Lemma 3. This proves (ii).*

*We now prove iii), that is*

$$\mathrm{Var}\left[L_t|\mathcal{F}_t\right] \le c(1 + \|\Xi_t\|^2). \tag{23}$$

*Now by equation 21 we have that*

$$\mathrm{Var}\left[L_t|\mathcal{F}_t\right] = \mathrm{Var}\left[\max\left\{\mathcal{M}^{\boldsymbol{\pi}}Q_2(s_{\tau_k},\boldsymbol{a}),\mathcal{R}^{\mathrm{re}}(s_{\tau_k},\boldsymbol{a}) + \gamma\max_{a'\in\mathcal{A}} Q_{2,t}(s',\boldsymbol{a}')\right\} - Q_2^\star(s_t,\boldsymbol{a})\right]$$

$$= \mathbb{E}\left[\left(\max\left\{\mathcal{M}^{\boldsymbol{\pi}}Q_2(s_{\tau_k},\boldsymbol{a}),\mathcal{R}^{\mathrm{re}}(s_{\tau_k},\boldsymbol{a}) + \gamma\max_{a'\in\mathcal{A}} Q_{2,t}(s',\boldsymbol{a}')\right\}\right.\right.$$

$$\left.\left. - Q_2^\star(s_t,a) - (TQ_{2,t}(s,\boldsymbol{a}) - Q_2^\star(s,\boldsymbol{a}))\right)^2\right]$$

$$= \mathbb{E}\left[\left(\max\left\{\mathcal{M}^{\boldsymbol{\pi}}Q_2(s_{\tau_k},\boldsymbol{a}),\mathcal{R}^{\mathrm{re}}(s_{\tau_k},\boldsymbol{a}) + \gamma\max_{a'\in\mathcal{A}} Q_{2,t}(s',\boldsymbol{a}')\right\} - TQ_{2,t}(s,\boldsymbol{a})\right)^2\right]$$

$$= \mathrm{Var}\left[\max\left\{\mathcal{M}^{\boldsymbol{\pi}}Q_2(s_{\tau_k},\boldsymbol{a}),\mathcal{R}^{\mathrm{re}}(s_{\tau_k},\boldsymbol{a}) + \gamma\max_{a'\in\mathcal{A}} Q_{2,t}(s',\boldsymbol{a}')\right\} - TQ_{2,t}(s,\boldsymbol{a}))^2\right]$$

$$\le c(1 + \|\Xi_t\|^2),$$

*for some $c > 0$ where the last line follows due to the boundedness of $Q_2$ (which follows from Assumptions 2 and 4).*

*This concludes the proof of Prop. 3.*

## Proof of Proposition 2

**Proof 5 (Proof of Prop. 2)** *The proof is given by establishing a contradiction. Therefore suppose that $\mathcal{M}^{\pi,\pi^{\mathrm{re}}}v_2^{\pi^{\mathrm{it}},\pi'^{\mathrm{re}}}(s_{\tau_k}) \le v_2^{\pi^{\mathrm{it}},\pi'^{\mathrm{re}}}(s_{\tau_k})$ and suppose that the intervention time $\tau_1' > \tau_1$ is an optimal intervention time. Construct the policy $\pi'^{\mathrm{re}} \in \Psi^{\mathrm{re}}$ and $\tilde{\pi}^{\mathrm{re}}$ policy switching times by $(\tau_0',\tau_1',\ldots,)$ and $\pi'^{\mathrm{re}} \in \Psi^{\mathrm{re}}$ policy by $(\tau_0',\tau_1,\ldots)$ respectively. Define by $l = \inf\{t > 0; \mathcal{M}^{\pi,\pi^{\mathrm{re}}}v_2^{\pi^{\mathrm{it}},\pi'^{\mathrm{re}}}(s_t) = v_2^{\pi^{\mathrm{it}},\pi'^{\mathrm{re}}}(s_t)\}$ and $m = \sup\{t; t < \tau_1'\}$. By construction we have that*

$$v_2^{\pi^{\mathrm{it}},\pi'^{\mathrm{re}}}(s)$$
$$= \mathbb{E}\left[L(s_0,a_0) + \mathbb{E}\left[\ldots + \gamma^{l-1}\mathbb{E}\left[L(s_{\tau_1-1},a_{\tau_1-1}) + \ldots + \gamma^{m-l-1}\mathbb{E}\left[L(s_{\tau_1'-1},a_{\tau_1'-1}) + \gamma\mathcal{M}^{\pi^{\mathrm{it}},\pi^{\mathrm{re}}}v_2^{\pi^{\mathrm{it}},\pi'^{\mathrm{re}}}(s')\right]\right]\right]\right]$$
$$< \mathbb{E}\left[L(s_0,a_0) + \mathbb{E}\left[\ldots + \gamma^{l-1}\mathbb{E}\left[L(s_{\tau_1-1},a_{\tau_1-1}) + \gamma\mathcal{M}^{\pi^{\mathrm{it}},\tilde{\pi}^{\mathrm{re}}}v_2^{\pi^{\mathrm{it}},\pi'^{\mathrm{re}}}(s_{\tau_1})\right]\right]\right]$$

*We now use the following observation $\mathbb{E}\left[L(s_{\tau_1-1},a_{\tau_1-1}) + \gamma\mathcal{M}^{\pi^{\mathrm{it}},\tilde{\pi}^{\mathrm{re}}}v_2^{\pi^{\mathrm{it}},\pi'^{\mathrm{re}}}(s_{\tau_1})\right]$*

$$\le \max\left\{\mathcal{M}^{\pi^{\mathrm{it}},\tilde{\pi}^{\mathrm{re}}}v_2^{\pi^{\mathrm{it}},\pi'^{\mathrm{re}}}(s_{\tau_1}), \max_{a_{\tau_1}\in\mathcal{A}}\left[L(s_{\tau_1},a_{\tau_1}) + \gamma\sum_{s'\in\mathcal{S}}\mathcal{P}(s';a_{\tau_1},s_{\tau_1})v_2^{\pi^{\mathrm{it}},\pi^{\mathrm{re}}}(s')\right]\right\}.$$

*Using this we deduce that*

$$v_2^{\pi^{\mathrm{it}},\pi'^{\mathrm{re}}}(s) \le \mathbb{E}\left[L(s_0,a_0) + \mathbb{E}\left[\ldots\right.\right.$$

$$\left.\left. + \gamma^{l-1}\mathbb{E}\left[-L(s_{\tau_1-1},a_{\tau_1-1}) + \gamma\max\left\{\mathcal{M}^{\pi^{\mathrm{it}},\tilde{\pi}^{\mathrm{re}}}v_2^{\pi^{\mathrm{it}},\pi'^{\mathrm{re}}}(s_{\tau_1}), \max_{a_{\tau_1}\in\mathcal{A}}\left[L(s_{\tau_1},a_{\tau_1}) + \gamma\sum_{s'\in\mathcal{S}}\mathcal{P}(s';a_{\tau_1},s_{\tau_1})v_2^{\pi^{\mathrm{it}},\pi^{\mathrm{re}}}(s')\right]\right\}\right]\right]\right]$$

$$= \mathbb{E}\left[L(s_0,a_0) + \mathbb{E}\left[\ldots + \gamma^{l-1}\mathbb{E}\left[L(s_{\tau_1-1},a_{\tau_1-1}) + \gamma\left[Tv_2^{\pi^{\mathrm{it}},\tilde{\pi}^{\mathrm{re}}}\right](s_{\tau_1})\right]\right]\right] = v_2^{\pi^{\mathrm{it}},\tilde{\pi}^{\mathrm{re}}}(s)),$$

*where the first inequality is true by assumption on $\mathcal{M}$. This is a contradiction since $\pi'^{\text{re}}$ is an optimal policy for Player 2. Using analogous reasoning, we deduce the same result for $\tau'_k < \tau_k$ after which deduce the result. Moreover, by invoking the same reasoning, we can conclude that it must be the case that $(\tau_0, \tau_1, \ldots, \tau_{k-1}, \tau_k, \tau_{k+1}, \ldots,)$ are the optimal switching times, this completes the proof of part (i).*

*We now prove part (ii). First, we note that it is easy to see that $v_2^{\pi^{\text{it}},(\pi^{\text{re}},\mathfrak{g})}$ is bounded above, indeed using the above we have that*

$$v_2^{\pi^{\text{it}},(\pi^{\text{re}},\mathfrak{g})}(s) = \mathbb{E}\left[\sum_{t\geq 0}\gamma^t\left(\mathcal{R}^{\text{re}}\left(s_t, a_t^{\text{it}}, a_t^{\text{re}}\right) - \beta g_t\right)\right] \tag{24}$$

$$= \mathbb{E}\left[\sum_{t=0}^{\infty}\gamma^t\left((L(s,a_t^{\text{re}})\mathbf{1}_{\mathcal{A}^{\text{re}}}(a_t^{\text{re}}) + L(s_t,a^{\text{it}})(1 - g_t - \beta g_t)\right)\right] \tag{25}$$

$$\leq \left|\mathbb{E}_{\pi,\pi^{\text{re}}}\left[\sum_{t=0}^{\infty}\gamma^t\left(2L - \beta g_t\right)\right]\right| \tag{26}$$

$$\leq \sum_{t=0}^{\infty}\gamma^t\left(2\|L\| + K\right) \tag{27}$$

$$= \frac{1}{1-\gamma}\left(2\|L\| + K\right), \tag{28}$$

*using the triangle inequality, the (upper-)boundedness of $L$ (Assumption 5). We now note that by the dominated convergence theorem we have that $\forall (s_0) \in \mathcal{S} \times \{0,1\}$*

$$\lim_{l\to 0} v_2^{\pi^{\text{it}},(\pi^{\text{re}},\mathfrak{g})}(s) = \lim_{l\to 0}\mathbb{E}\left[\sum_{t\geq 0}\gamma^t\left(\mathcal{R}^{\text{re}}\left(s_t, a_t^{\text{it}}, a_t^{\text{re}}\right) - \beta g_t\right)\right] \tag{29}$$

$$= \lim_{l\to 0}\mathbb{E}\left[\sum_{t=0}^{\infty}\gamma^t\left(L(s,a_t^{\text{re}})g_t + L(s_t,a^{\text{it}})(1 - g_t) - \beta g_t\right)\right] \tag{30}$$

$$= \mathbb{E}\left[\lim_{l\to 0}\sum_{t=0}^{\infty}\gamma^t\left(L(s,a_t^{\text{re}})g_t + L(s_t,a^{\text{it}})(1 - g_t) - \beta g_t\right)\right] \tag{31}$$

$$= -\beta\mathbb{E}\left[\sum_{t=0}^{\infty}\gamma^t g_t\right], \tag{32}$$

*using Assumption 6 in the last step, after which we deduce (ii) since equation 32 is maximised when $g_t = 0$ for all $t = 0, 1, \ldots$ which is achieved only when $\mu_l(\mathfrak{g}) = 0$. Additionally, by part (i) we have that*

$$\tau_k = \inf\left\{\tau > \tau_{k-1}|\mathcal{M}^{\Psi^{\text{re}}}v_2^{\pi^{\text{it}},\Psi^{\text{re}}} = v_2^{\pi^{\text{it}},\Psi^{\text{re}}}\right\}. \tag{33}$$

*It is easy to see that given equation 32 and the definition of $\mathcal{M}$ (c.f. Proposition 3), condition equation 33 can never be satisfied which implies that **Switcher** performs no interventions.*

*This completes the proof of Prop. 2.*

*To complete the proof of Theorem 1, we prove the following result:*

**Lemma 4** *The **Explorer** learns to solve the MDP $\langle \mathcal{S}, \mathcal{A}, P, R, \gamma \rangle$ and its value function converges.*

**Proof 6** *We first deduce the boundedness of the **Exploiter** objective $v_1^{\pi^{\text{it}},(\pi^{\text{re}},\mathfrak{g})}$:*

$$v_1^{\pi^{\text{it}},(\pi^{\text{re}},\mathfrak{g})}(s) = \mathbb{E}\left[\sum_{t\geq 0}\gamma_1^t \mathcal{R}^{\text{it}}\left(s_t, a_t^{\text{it}}, a_t^{\text{re}}\right)\right]$$

$$= \mathbb{E}\left[\sum_{t\geq 0}\gamma_1^t\left(\mathcal{R}(s_t, a_t^{\text{it}})(1-g_t) + \mathcal{R}(s, a_t^{\text{re}})g_t\right)\right]$$

$$\leq \mathbb{E}\left[\sum_{t\geq 0}\gamma_1^t\left(\mathcal{R}(s_t, a_t^{\text{it}}) + \mathcal{R}(s, a_t^{\text{re}})\right)\right]$$

$$\leq \frac{2}{1-\gamma_1}\|\mathcal{R}\|,$$

*therefore the **Explorer**'s objective is bounded above by some finite quantity.*

*Using the kronecker-delta function, the **Exploiter** objective as:*

$$v_1^{\pi^{\text{it}},(\pi^{\text{re}},\mathfrak{g})}(s) = \mathbb{E}\left[\sum_{t\geq 0}\sum_{k=0}^{\mu_l(\mathfrak{g})}\gamma_1^t \mathcal{R}(s_t, a_t^{\text{it}})(1-\delta_{\tau_k}^t) + \mathcal{R}(s, a_t^{\text{re}})\delta_{\tau_k}^t\right]. \tag{34}$$

*Recall that $\mu_l(\mathfrak{g})$ denotes the number of switch activations performed by the **Switcher**. Denote by $\mathfrak{g}_0$ the **Explorer** intervention policy that performs no interventions. By Prop. 2 and by the dominated convergence theorem, we have that $\forall s \in \mathcal{S}$*

$$\lim_{l\to 0}v_1^{\pi^{\text{it}},(\pi^{\text{re}},\mathfrak{g})}(s) = \lim_{l\to 0}\mathbb{E}_{P,\pi^{\text{it}},(\pi^{\text{re}},\mathfrak{g})}\left[\sum_{t\geq 0}\sum_{k=0}^{\mu_l(\mathfrak{g})}\gamma_1^t \mathcal{R}(s_t, a_t^{\text{it}})(1-\delta_{\tau_k}^t) + \mathcal{R}(s, a_t^{\text{re}})\delta_{\tau_k}^t\right]$$

$$= \mathbb{E}_{P,\pi^{\text{it}},(\pi^{\text{re}},\mathfrak{g})}\lim_{l\to 0}\left[\sum_{t\geq 0}\sum_{k=0}^{\mu_l(\mathfrak{g})}\gamma_1^t \mathcal{R}(s_t, a_t^{\text{it}})(1-\delta_{\tau_k}^t) + \mathcal{R}(s, a_t^{\text{re}})\delta_{\tau_k}^t\right]$$

$$= \mathbb{E}_{P,\pi^{\text{it}},(\pi^{\text{re}},\mathfrak{g}_0)}\left[\sum_{t\geq 0}\gamma_1^t \mathcal{R}(s_t, a_t^{\text{it}})\right]$$

$$= \mathbb{E}_{P,\pi^{\text{it}}}\left[\sum_{t\geq 0}\gamma_1^t \mathcal{R}(s_t, a_t^{\text{it}})\right] = v_1^{\pi^{\text{it}}}(s),$$

*using the fact that $\lim_{l\to 0}\mu_l = 0$ and by Fubini's theorem in the penultimate step. Therefore, in the limit $l\to 0$, the **Exploiter** solves the MDP $\langle \mathcal{S}, \mathcal{A}, P, R, \gamma\rangle$ which converges to a stable point.*

**Corollary 1** *After combining Lemma 4, Prop. 2 and Prop. 3 we deduce the result of Theorem 1.*

## Proof of Convergence with Function Approximation

*First let us recall the statement of the theorem:*

**Theorem 3** *SEREN converges to a limit point $r^\star$ which is the unique solution to the equation:*

$$\Pi\mathfrak{F}(\Phi r^\star) = \Phi r^\star, \qquad a.e. \tag{35}$$

*where we recall that for any test function $\Lambda \in \mathcal{V}$, the operator $\mathfrak{F}$ is defined by $\mathfrak{F}\Lambda := \Theta + \gamma P \max\{\mathcal{M}\Lambda, \Lambda\}$.*

*Moreover, $r^\star$ satisfies the following:*

$$\|\Phi r^\star - Q_2^\star\| \leq c\|\Pi Q_2^\star - Q_2^\star\|. \tag{36}$$

The theorem is proven using a set of results that we now establish. To this end, we first wish to prove the following bound:

**Lemma 5** For any $Q \in \mathcal{V}$ we have that

$$\|\mathfrak{F}Q_2 - Q_2'\| \leq \gamma \|Q_2 - Q_2'\|, \tag{37}$$

so that the operator $\mathfrak{F}$ is a contraction.

**Proof 7** Recall, for any test function $\psi$, a projection operator $\Pi$ acting $\Lambda$ is defined by the following

$$\Pi\Lambda := \underset{\bar{\Lambda} \in \{\Phi r | r \in \mathbb{R}^p\}}{\arg\min} \|\bar{\Lambda} - \Lambda\|.$$

Now, we first note that in the proof of Lemma 3, we deduced that for any $\Lambda \in L_2$ we have that

$$\left\|\mathcal{M}^{\boldsymbol{\pi}}\Lambda - \left[\mathcal{R}(\cdot, \boldsymbol{a}) + \gamma\underset{\boldsymbol{a} \in \mathcal{A}}{\max} \mathcal{P}^{\boldsymbol{a}}\Lambda'\right]\right\| \leq \gamma \|\Lambda - \Lambda'\|,$$

(c.f. Lemma 3).

Setting $\Lambda = Q_2$ and $\psi = \mathcal{R}$, it can be straightforwardly deduced that for any $Q_2, \hat{Q}_2 \in L_2$: $\left\|\mathcal{M}^{\boldsymbol{\pi}}Q_2 - \hat{Q}_2\right\| \leq \gamma \left\|Q_2 - \hat{Q}_2\right\|$. Hence, using the contraction property of $\mathcal{M}$, we readily deduce the following bound:

$$\max\left\{\left\|\mathcal{M}^{\boldsymbol{\pi}}Q_2 - \hat{Q}_2\right\|, \left\|\mathcal{M}^{\boldsymbol{\pi}}Q_2 - \mathcal{M}\hat{Q}_2\right\|\right\} \leq \gamma \left\|Q_2 - \hat{Q}_2\right\|, \tag{38}$$

We now observe that $\mathfrak{F}$ is a contraction. Indeed, since for any $Q_2, Q_2' \in L_2$ we have that:

$$\begin{aligned}
\|\mathfrak{F}Q_2 - \mathfrak{F}Q_2'\| &= \|\Theta + \gamma P \max\{\mathcal{M}^{\boldsymbol{\pi}}Q_2, Q_2\} - (\Theta + \gamma P \max\{\mathcal{M}^{\boldsymbol{\pi}}Q_2', Q_2'\})\| \\
&= \gamma \|P \max\{\mathcal{M}^{\boldsymbol{\pi}}Q_2, Q_2\} - P \max\{\mathcal{M}^{\boldsymbol{\pi}}Q_2', Q_2'\}\| \\
&\leq \gamma \|\max\{\mathcal{M}^{\boldsymbol{\pi}}Q_2, Q_2\} - \max\{\mathcal{M}^{\boldsymbol{\pi}}Q_2', Q_2'\}\| \\
&\leq \gamma \|\max\{\mathcal{M}^{\boldsymbol{\pi}}Q_2 - \mathcal{M}^{\boldsymbol{\pi}}Q_2', Q_2 - \mathcal{M}^{\boldsymbol{\pi}}Q_2', \mathcal{M}^{\boldsymbol{\pi}}Q_2 - Q_2', Q_2 - Q_2'\}\| \\
&\leq \gamma \max\{\|\mathcal{M}^{\boldsymbol{\pi}}Q_2 - \mathcal{M}^{\boldsymbol{\pi}}Q_2'\|, \|Q_2 - \mathcal{M}^{\boldsymbol{\pi}}Q_2'\|, \|\mathcal{M}^{\boldsymbol{\pi}}Q_2 - Q_2'\|, \|Q_2 - Q_2'\|\} \\
&= \gamma \|Q_2 - Q_2'\|,
\end{aligned}$$

using equation 38 and again using the non-expansiveness of $P$.

We next show that the following two bounds hold:

**Lemma 6** For any $Q_2 \in \mathcal{V}$ we have that

> i) $\qquad \left\|\Pi\mathfrak{F}Q_2 - \Pi\mathfrak{F}\bar{Q}_2\right\| \leq \gamma \left\|Q_2 - \bar{Q}_2\right\|,$
>
> ii) $\qquad \|\Phi r^\star - Q_2^\star\| \leq \frac{1}{\sqrt{1-\gamma^2}} \|\Pi Q_2^\star - Q_2^\star\|.$

**Proof 8** The first result is straightforward since as $\Pi$ is a projection it is non-expansive and hence:

$$\left\|\Pi\mathfrak{F}Q_2 - \Pi\mathfrak{F}\bar{Q}_2\right\| \leq \left\|\mathfrak{F}Q_2 - \mathfrak{F}\bar{Q}_2\right\| \leq \gamma \left\|Q_2 - \bar{Q}_2\right\|,$$

using the contraction property of $\mathfrak{F}$. This proves i). For ii), we note that by the orthogonality property of projections we have that $\langle \Phi r^\star - \Pi Q_2^\star, \Phi r^\star - \Pi Q_2^\star \rangle$, hence we observe that:

$$\begin{aligned}
\|\Phi r^\star - Q_2^\star\|^2 &= \|\Phi r^\star - \Pi Q_2^\star\|^2 + \|\Phi r^\star - \Pi Q_2^\star\|^2 \\
&= \|\Pi\mathfrak{F}\Phi r^\star - \Pi Q_2^\star\|^2 + \|\Phi r^\star - \Pi Q_2^\star\|^2 \\
&\leq \|\mathfrak{F}\Phi r^\star - Q_2^\star\|^2 + \|\Phi r^\star - \Pi Q_2^\star\|^2 \\
&= \|\mathfrak{F}\Phi r^\star - \mathfrak{F}Q_2^\star\|^2 + \|\Phi r^\star - \Pi Q_2^\star\|^2 \\
&\leq \gamma^2 \|\Phi r^\star - Q_2^\star\|^2 + \|\Phi r^\star - \Pi Q_2^\star\|^2,
\end{aligned}$$

after which we readily deduce the desired result.

**Lemma 7** *Define the operator $H$ by the following:* $HQ_2(z) = \begin{cases} \mathcal{M}^{\boldsymbol{\pi}} Q_2(z), & \text{if } \mathcal{M}^{\boldsymbol{\pi}} Q_2(s) > \Phi r^\star, \\ Q_2(z), & \text{otherwise}, \end{cases}$

*and $\tilde{\mathfrak{F}}$ by:* $\tilde{\mathfrak{F}} Q_2 := \mathcal{R} + \gamma P H Q_2.$

*For any $Q_2, \bar{Q}_2 \in L_2$ we have that*

$$\left\| \tilde{\mathfrak{F}} Q_2 - \tilde{\mathfrak{F}} \bar{Q}_2 \right\| \leq \gamma \left\| Q_2 - \bar{Q}_2 \right\| \tag{39}$$

*and hence $\tilde{\mathfrak{F}}$ is a contraction mapping.*

**Proof 9** *Using equation 38, we now observe that*

$$\begin{aligned}
\left\| \tilde{\mathfrak{F}} Q_2 - \tilde{\mathfrak{F}} \bar{Q}_2 \right\| &= \left\| \mathcal{R} + \gamma P H Q_2 - \left( \mathcal{R} + \gamma P H \bar{Q}_2 \right) \right\| \\
&\leq \gamma \left\| H Q_2 - H \bar{Q}_2 \right\| \\
&\leq \gamma \left\| \max \left\{ \mathcal{M}^{\boldsymbol{\pi}} Q_2 - \mathcal{M} \bar{Q}_2, Q_2 - \bar{Q}_2, \mathcal{M}^{\boldsymbol{\pi}} Q_2 - \bar{Q}_2, \mathcal{M} \bar{Q}_2 - Q_2 \right\} \right\| \\
&\leq \gamma \max \left\{ \left\| \mathcal{M}^{\boldsymbol{\pi}} Q_2 - \mathcal{M} \bar{Q}_2 \right\|, \left\| Q_2 - \bar{Q}_2 \right\|, \left\| \mathcal{M}^{\boldsymbol{\pi}} Q_2 - \bar{Q}_2 \right\|, \left\| \mathcal{M} \bar{Q}_2 - Q_2 \right\| \right\} \\
&\leq \gamma \max \left\{ \gamma \left\| Q_2 - \bar{Q}_2 \right\|, \left\| Q_2 - \bar{Q}_2 \right\|, \left\| \mathcal{M}^{\boldsymbol{\pi}} Q_2 - \bar{Q}_2 \right\|, \left\| \mathcal{M} \bar{Q}_2 - Q_2 \right\| \right\} \\
&= \gamma \left\| Q_2 - \bar{Q}_2 \right\|,
\end{aligned}$$

*again using the non-expansive property of $P$.*

**Lemma 8** *Define by $\tilde{Q}_2 := \mathcal{R}^{\mathrm{re}} + \gamma P v_2^{\tilde{\boldsymbol{\pi}}}$ where*

$$v_2^{\tilde{\boldsymbol{\pi}}}(s) := \mathcal{R}_2(s_{\tau_k}, a) + \gamma \max_{a \in \mathcal{A}} \sum_{s' \in \mathcal{S}} \mathcal{P}(s'; a, s_{\tau_k}) \Phi r^\star(s'), \tag{40}$$

*then $\tilde{Q}_2$ is a fixed point of $\tilde{\mathfrak{F}} \tilde{Q}_2$, that is $\tilde{\mathfrak{F}} \tilde{Q}_2 = \tilde{Q}_2$.*

**Proof 10** *We begin by observing that*

$$\begin{aligned}
H \tilde{Q}_2(z) &= H \left( L(z) + \gamma P v_2^{\tilde{\boldsymbol{\pi}}} \right) \\
&= \begin{cases} \mathcal{M}^{\boldsymbol{\pi}} Q_2(z), & \text{if } \mathcal{M}^{\boldsymbol{\pi}} Q_2(z) > \Phi r^\star, \\ Q_2(z), & \text{otherwise}, \end{cases} \\
&= \begin{cases} \mathcal{M}^{\boldsymbol{\pi}} Q_2(z), & \text{if } \mathcal{M}^{\boldsymbol{\pi}} Q_2(z) > \Phi r^\star, \\ L(z) + \gamma P v_2^{\tilde{\boldsymbol{\pi}}}, & \text{otherwise}, \end{cases} \\
&= v_2^{\tilde{\boldsymbol{\pi}}}(s).
\end{aligned}$$

*Hence,*

$$\tilde{\mathfrak{F}} \tilde{Q}_2 = \mathcal{R} + \gamma P H \tilde{Q}_2 = \mathcal{R} + \gamma P v_2^{\tilde{\boldsymbol{\pi}}} = \tilde{Q}_2. \tag{41}$$

*which proves the result.*

**Lemma 9** *The following bound holds:*

$$\mathbb{E}\left[ v_2^{\hat{\boldsymbol{\pi}}}(s_0) \right] - \mathbb{E}\left[ v_2^{\tilde{\boldsymbol{\pi}}}(s_0) \right] \leq 2 \left[ (1 - \gamma) \sqrt{(1 - \gamma^2)} \right]^{-1} \left\| \Pi Q_2^\star - Q_2^\star \right\|. \tag{42}$$

**Proof 11** *By definitions of $v_2^{\hat{\boldsymbol{\pi}}}$ and $v_2^{\tilde{\boldsymbol{\pi}}}$ (c.f equation 40) and using Jensen's inequality and the stationarity property we have that,*

$$\begin{aligned}
\mathbb{E}\left[ v_2^{\hat{\boldsymbol{\pi}}}(s_0) \right] - \mathbb{E}\left[ v_2^{\tilde{\boldsymbol{\pi}}}(s_0) \right] &= \mathbb{E}\left[ P v_2^{\hat{\boldsymbol{\pi}}}(z_0) \right] - \mathbb{E}\left[ P v_2^{\tilde{\boldsymbol{\pi}}}(z_0) \right] \\
&\leq \left| \mathbb{E}\left[ P v_2^{\hat{\boldsymbol{\pi}}}(z_0) \right] - \mathbb{E}\left[ P v_2^{\tilde{\boldsymbol{\pi}}}(z_0) \right] \right| \\
&\leq \left\| P v_2^{\hat{\boldsymbol{\pi}}} - P v_2^{\tilde{\boldsymbol{\pi}}} \right\|. \tag{43}
\end{aligned}$$

Now recall that $\tilde{Q}_2 := \mathcal{R} + \gamma P v_2^{\tilde{\pi}}$ and $Q_2^\star := \mathcal{R} + \gamma P v_2^{\pi^\star}$, using these expressions in equation 43 we find that

$$\mathbb{E}\left[v_2^{\hat{\pi}}(z_0)\right] - \mathbb{E}\left[v_2^{\tilde{\pi}}(z_0)\right] \leq \frac{1}{\gamma}\left\|\tilde{Q}_2 - Q_2^\star\right\|.$$

Moreover, by the triangle inequality and using the fact that $\mathfrak{F}(\Phi r^\star) = \tilde{\mathfrak{F}}(\Phi r^\star)$ and that $\mathfrak{F}Q_2^\star = Q_2^\star$ and $\mathfrak{F}\tilde{Q}_2 = \tilde{Q}_2$ (c.f. equation 42) we have that

$$\begin{aligned}
\left\|\tilde{Q}_2 - Q_2^\star\right\| &\leq \left\|\tilde{Q}_2 - \mathfrak{F}(\Phi r^\star)\right\| + \left\|Q_2^\star - \tilde{\mathfrak{F}}(\Phi r^\star)\right\| \\
&\leq \gamma\left\|\tilde{Q}_2 - \Phi r^\star\right\| + \gamma\left\|Q_2^\star - \Phi r^\star\right\| \\
&\leq 2\gamma\left\|\tilde{Q}_2 - \Phi r^\star\right\| + \gamma\left\|Q_2^\star - \tilde{Q}_2\right\|,
\end{aligned}$$

which gives the following bound:

$$\left\|\tilde{Q}_2 - Q_2^\star\right\| \leq 2\left(1-\gamma\right)^{-1}\left\|\tilde{Q}_2 - \Phi r^\star\right\|,$$

from which, using Lemma 6, we deduce that $\left\|\tilde{Q}_2 - Q_2^\star\right\| \leq 2\left[(1-\gamma)\sqrt{(1-\gamma^2)}\right]^{-1}\left\|\tilde{Q}_2 - \Phi r^\star\right\|$, after which by equation 44, we finally obtain

$$\mathbb{E}\left[v_2^{\hat{\pi}}(s_0)\right] - \mathbb{E}\left[v_2^{\tilde{\pi}}(s_0)\right] \leq 2\left[(1-\gamma)\sqrt{(1-\gamma^2)}\right]^{-1}\left\|\tilde{Q}_2 - \Phi r^\star\right\|,$$

as required.

Let us rewrite the update in the following way:

$$r_{t+1} = r_t + \gamma_t \Xi_2(w_t, r_t),$$

where the function $\Xi_2 : \mathbb{R}^{2d} \times \mathbb{R}^p \to \mathbb{R}^p$ is given by:

$$\Xi_2(w, r) := \phi(z)\left(L(z) + \gamma \max\left\{(\Phi r)(z'), \mathcal{M}(\Phi r)(z')\right\} - (\Phi r)(s)\right),$$

for any $w \equiv (z, z') \in (\mathbb{N} \times \mathcal{S})^2$ where $z = (t, s) \in \mathbb{N} \times \mathcal{S}$ and $z' = (t, s') \in \mathbb{N} \times \mathcal{S}$ and for any $r \in \mathbb{R}^p$. Let us also define the function $\mathbf{\Xi}_2 : \mathbb{R}^p \to \mathbb{R}^p$ by the following:

$$\mathbf{\Xi}_2(r) := \mathbb{E}_{w_0 \sim (\mathbb{P}, \mathbb{P})}\left[\Xi_2(w_0, r)\right]; w_0 := (z_0, z_1).$$

**Lemma 10** The following statements hold for all $z \in \{0, 1\} \times \mathcal{S}$:

    i) $(r - r^\star)\mathbf{\Xi}_{2,k}(r) < 0, \qquad \forall r \neq r^\star,$

    ii) $\mathbf{\Xi}_{2,k}(r^\star) = 0.$

**Proof 12** To prove the statement, we first note that each component of $\mathbf{\Xi}_{2,k}(r)$ admits a representation as an inner product, indeed:

$$\begin{aligned}
\mathbf{\Xi}_{2,k}(r) &= \mathbb{E}\left[\phi_k(z_0)(L(z_0) + \gamma \max\left\{\Phi r(z_1), \mathcal{M}^{\boldsymbol{\pi}}\Phi(z_1)\right\} - (\Phi r)(z_0)\right] \\
&= \mathbb{E}\left[\phi_k(z_0)(L(z_0) + \gamma \mathbb{E}\left[\max\left\{\Phi r(z_1), \mathcal{M}^{\boldsymbol{\pi}}\Phi(z_1)\right\}|z_0\right] - (\Phi r)(z_0)\right] \\
&= \mathbb{E}\left[\phi_k(z_0)(L(z_0) + \gamma P \max\left\{(\Phi r, \mathcal{M}^{\boldsymbol{\pi}}\Phi)\right\}(z_0) - (\Phi r)(z_0)\right] \\
&= \left\langle \phi_k, \mathfrak{F}\Phi r - \Phi r\right\rangle,
\end{aligned}$$

using the iterated law of expectations and the definitions of $P$ and $\mathfrak{F}$.

We now are in position to prove i). Indeed, we now observe the following:

$$\begin{aligned}
(r - r^\star)\mathbf{\Xi}_{2,k}(r) &= \sum_{l=1}\left(r(l) - r^\star(l)\right)\left\langle\phi_l, \mathfrak{F}\Phi r - \Phi r\right\rangle \\
&= \left\langle\Phi r - \Phi r^\star, \mathfrak{F}\Phi r - \Phi r\right\rangle \\
&= \left\langle\Phi r - \Phi r^\star, (\mathbf{1} - \Pi)\mathfrak{F}\Phi r + \Pi\mathfrak{F}\Phi r - \Phi r\right\rangle \\
&= \left\langle\Phi r - \Phi r^\star, \Pi\mathfrak{F}\Phi r - \Phi r\right\rangle,
\end{aligned}$$

*where in the last step we used the orthogonality of $(\mathbf{1} - \Pi)$. We now recall that $\Pi\mathfrak{F}\Phi r^\star = \Phi r^\star$ since $\Phi r^\star$ is a fixed point of $\Pi\mathfrak{F}$. Additionally, using Lemma 6 we observe that $\|\Pi\mathfrak{F}\Phi r - \Phi r^\star\| \le \gamma\|\Phi r - \Phi r^\star\|$. With this we now find that*

$$\langle \Phi r - \Phi r^\star, \Pi\mathfrak{F}\Phi r - \Phi r\rangle$$
$$= \langle \Phi r - \Phi r^\star, (\Pi\mathfrak{F}\Phi r - \Phi r^\star) + \Phi r^\star - \Phi r\rangle$$
$$\le \|\Phi r - \Phi r^\star\| \|\Pi\mathfrak{F}\Phi r - \Phi r^\star\| - \|\Phi r^\star - \Phi r\|^2$$
$$\le (\gamma - 1) \|\Phi r^\star - \Phi r\|^2,$$

*which is negative since $\gamma < 1$ which completes the proof of part i).*

*The proof of part ii) is straightforward since we readily observe that*

$$\Xi_{2,k}(r^\star) = \langle \phi_l, \mathfrak{F}\Phi r^\star - \Phi r\rangle = \langle \phi_l, \Pi\mathfrak{F}\Phi r^\star - \Phi r\rangle = 0,$$

*as required and from which we deduce the result.*

*To prove the theorem, we make use of a special case of the following result:*

**Theorem 3 (Th. 17, p. 239 in Benveniste et al. (2012))** *Consider a stochastic process $r_t : \mathbb{R} \times \{\infty\} \times \Omega \to \mathbb{R}^k$ which takes an initial value $r_0$ and evolves according to the following:*

$$r_{t+1} = r_t + \alpha\Xi_2(s_t, r_t), \tag{44}$$

*for some function $s : \mathbb{R}^{2d} \times \mathbb{R}^k \to \mathbb{R}^k$ and where the following statements hold:*

1. *$\{s_t | t = 0, 1, \ldots\}$ is a stationary, ergodic Markov process taking values in $\mathbb{R}^{2d}$*

2. *For any positive scalar $q$, there exists a scalar $\mu_q$ such that $\mathbb{E}\left[1 + \|s_t\|^q | s \equiv s_0\right] \le \mu_q\left(1 + \|s\|^q\right)$*

3. *The step size sequence satisfies the Robbins-Monro conditions, that is $\sum_{t=0}^\infty \alpha_t = \infty$ and $\sum_{t=0}^\infty \alpha_t^2 < \infty$*

4. *There exists scalars $c$ and $q$ such that $\|\Xi_2(w, r)\| \le c\left(1 + \|w\|^q\right)\left(1 + \|r\|\right)$*

5. *There exists scalars $c$ and $q$ such that $\sum_{t=0}^\infty \|\mathbb{E}\left[\Xi_2(w_t, r) | z_0 \equiv z\right] - \mathbb{E}\left[\Xi_2(w_0, r)\right]\| \le c\left(1 + \|w\|^q\right)\left(1 + \|r\|\right)$*

6. *There exists a scalar $c > 0$ such that $\|\mathbb{E}[\Xi_2(w_0, r)] - \mathbb{E}[\Xi_2(w_0, \bar{r})]\| \le c\|r - \bar{r}\|$*

7. *There exists scalars $c > 0$ and $q > 0$ such that $\sum_{t=0}^\infty \|\mathbb{E}\left[\Xi_2(w_t, r) | w_0 \equiv w\right] - \mathbb{E}\left[\Xi_2(w_0, \bar{r})\right]\| \le c\|r - \bar{r}\|\left(1 + \|w\|^q\right)$*

8. *There exists some $r^\star \in \mathbb{R}^k$ such that $\bar{\Xi}_2(r)(r - r^\star) < 0$ for all $r \ne r^\star$ and $\bar{s}(r^\star) = 0$.*

*Then $r_t$ converges to $r^\star$ almost surely.*

*In order to apply the Theorem 3, we show that conditions 1 - 7 are satisfied.*

*Conditions 1-2 are true by assumption while condition 3 can be made true by choice of the learning rates. Therefore it remains to verify conditions 4-7 are met.*

*To prove 4, we observe that*

$$\|\Xi_2(w, r)\| = \|\phi(s)\left(L(z) + \gamma\max\left\{(\Phi r)(z'), \mathcal{M}^{\boldsymbol{\pi}}\Phi(z')\right\} - (\Phi r)(z)\right)\|$$
$$\le \|\phi(z)\| \|L(z) + \gamma\left(\|\phi(z')\| \|r\| + \mathcal{M}^{\boldsymbol{\pi}}\Phi(z')\right)\| + \|\phi(z)\| \|r\|$$
$$\le \|\phi(z)\|\left(\|L(z)\| + \gamma\|\mathcal{M}^{\boldsymbol{\pi}}\Phi(z')\|\right) + \|\phi(z)\|\left(\gamma\|\phi(z')\| + \|\phi(z)\|\right)\|r\|.$$

*Now using the definition of $\mathcal{M}$, we readily observe that $\|\mathcal{M}^{\boldsymbol{\pi}}\Phi(z')\| \le \|\mathcal{R}\| + \gamma\|\mathcal{P}_{s's_t}^{\pi}\Phi\| \le \|\mathcal{R}\| + \gamma\|\Phi\|$ using the non-expansiveness of $P$.*

*Hence, we lastly deduce that*

$$\|\Xi_2(w,r)\| \le \|\phi(z)\|\left(\|L(z)\| + \gamma\|\mathcal{M}^{\boldsymbol{\pi}}\Phi(z')\|\right) + \|\phi(z)\|\left(\gamma\|\phi(z')\| + \|\phi(z)\|\right)\|r\|$$
$$\le \|\phi(z)\|\left(\|L(z)\| + \gamma\|\mathcal{R}\| + \gamma\|\psi\|\right) + \|\phi(z)\|\left(\gamma\|\phi(z')\| + \|\phi(z)\|\right)\|r\|,$$

*we then easily deduce the result using the boundedness of $\phi, \mathcal{R}$ and $\psi$.*

*Now we observe the following Lipschitz condition on $\Xi_2$:*

$$\|\Xi_2(w,r) - \Xi_2(w,\bar{r})\|$$
$$= \|\phi(z)\left(\gamma\max\{(\Phi r)(z'), \mathcal{M}^{\boldsymbol{\pi}}\Phi(z')\} - \gamma\max\{(\Phi\bar{r})(z'), \mathcal{M}^{\boldsymbol{\pi}}\Phi(z')\}\right) - ((\Phi r)(z) - \Phi\bar{r}(z))\|$$
$$\le \gamma\|\phi(z)\|\|\max\{\phi'(z')r, \mathcal{M}^{\boldsymbol{\pi}}\Phi'(z')\} - \max\{(\phi'(z')\bar{r}), \mathcal{M}^{\boldsymbol{\pi}}\Phi'(z')\}\| + \|\phi(z)\|\|\phi'(z)r - \phi(z)\bar{r}\|$$
$$\le \gamma\|\phi(z)\|\|\phi'(z')r - \phi'(z')\bar{r}\| + \|\phi(z)\|\|\phi'(z)r - \phi'(z)\bar{r}\|$$
$$\le \|\phi(z)\|\left(\|\phi(z)\| + \gamma\|\phi(z)\|\|\phi'(z') - \phi'(z')\|\right)\|r - \bar{r}\|$$
$$\le c\|r - \bar{r}\|,$$

*using Cauchy-Schwarz inequality and that for any scalars $a, b, c$ we have that $|\max\{a,b\} - \max\{b,c\}| \le |a - c|$.*

*Using Assumptions 3 and 4, we therefore deduce that*

$$\sum_{t=0}^{\infty}\|\mathbb{E}\left[\Xi_2(w,r) - \Xi_2(w,\bar{r})|w_0 = w\right] - \mathbb{E}\left[\Xi_2(w_0,r) - \Xi_2(w_0,\bar{r})|\right]\| \le c\|r - \bar{r}\|(1 + \|w\|^l). \tag{45}$$

*Part 2 is assured by Lemma 6 while Part 4 is assured by Lemma 9 and lastly Part 8 is assured by Lemma 10.*

To complete the proof of Theorem 1, we make use of Theorem 1.1. in Borkar (1997) in which case we readily verify that with the appropriate choices of timesteps the Theorem is readily satisfied.

## C  Alternative uncertainty measures

● **Model-Based Ensemble Disagreement.**  By employing an ensemble of dynamics models, $\{\mathcal{M}_1, \ldots, \mathcal{M}_M\}$, where each $\mathcal{M}_m \in \mathcal{F}$ for $m = 1, \ldots, M$. Model training entails independent training of each of the model in the ensemble with the identical objectives (e.g., minimising the L2 distance between the predicted and the ground-truth next states). The uncertainty about a state-action pair $(s, a)$, can be quantified as the predictive ensemble disagreement:

$$L(s,a) = \frac{1}{M-1}\sum_{m=1}^{M}(\mathcal{M}_m(s,a) - \mathcal{E}_{\mathcal{M}}(s,a)) \tag{46}$$

where $\mathcal{E}_{\mathcal{M}}(s,a) = \frac{1}{M}\sum_{m=1}^{M}\mathcal{M}_e(s,a)$ is the empirical mean of the ensemble predictions. This approach has a information-theoretic interpretation such that through training, the mutual information between the dynamics model parameters and next-state is maximised, hence relating the epistemic uncertainty with the information-theoretic framework.

● **Integrating control into dynamics modelling with LSSM.** Consider we embed the dynamics modelling problem into a sequential modelling problem using latent state-space models (LSSM), using amortised inference, we are able to achieve fast inference and learning of the probabilistic graphical model. We could additionally incorporate action into the LSSM as a global factor that (potentially) influences both the latent

and observable codes. For instance, the generative process could be modelled as:

$$p(\mathbf{x}_{1:T}, \mathbf{z}_{1:T}, \mathbf{a}_{1:T-1}) = p(z_1)p(x_1|z_1) \cdot$$

$$\cdot \prod_{t=2}^{T} p(z_t|z_{t-1}, a_{t-1})p(x_t|z_t)p(a_{t-1}|z_{t-1}, x_{t-1}).$$

We could easily train an LSSM by maximising the variational lower bound utilising amortised inference. In the meantime, we could quantify the model uncertainty about the state-action pair $(z_t, a_t)$ in terms of the variance of the latent predictive distributions. By random trajectory-sampling (multiple particles), we target regions of the action space that maximises the predictive variance (assuming Gaussian for now). This could be achieved by importance-weighting on the computation of the marginal variance. Hence in this case we use the following uncertainty instantiation:

$$L(s, a) = \mathbb{V}(s'|s, a). \tag{47}$$

We could consider equation 47 as a parametric generalisation of equation 46 (despite the fact that in the model-ensemble method, the action is taken as an external input instead of a random variable as in the LSSM method).

## D    Implementation Details

The reinforcement learning agents (both SEREN and related baselines) are implemented in PyTorch (Paszke et al., 2019), and our implementations of baseline agents (SAC) are based on Stable-Baselines 3 (Raffin et al., 2021).

### D.1    MuJoCo Tasks

For SEREN-SAC, we use MLPs as the function approximator, with Adam optimiser (Kingma & Ba, 2014). We show the implementation details of the SEREN-SAC agent in Table 2 that are used in all studied MuJoCo environments. We note that the baseline SAC agent is implemented using exactly the same hyperparameters as Exploiter agent in SEREN-SAC.

### D.2    Atari 100K Benchmarks

For the Atari 100K Benchmarks, we use the Efficient-DQN structure as the baseline algorithm (Kostrikov et al., 2020), which augments a standard DQN with double Q-learning (Van Hasselt et al., 2016), dueling network for value estimation (Wang et al., 2016), and multi-step return as the TD target (Mnih et al., 2016). We compare our model, SEREN-Eff-DQN, against the following baseline algorithms: Rainbow (Hessel et al., 2018), Data-Efficient Rainbow (Efficient-Rainbow; Van Hasselt et al. (2019)), Efficient-DQN, DrQ (Kostrikov et al., 2020). The specific implementation details of SEREN-Eff-DQN can be found in Table 3.

We compute the human normalised score as human_normaliased_score $= \frac{\text{agent\_score} - \text{random\_score}}{\text{human\_score} - \text{random\_score}}$ (Mnih et al., 2013). The evaluation is based on $1.25 \times 10^5$ steps at the end of $1 \times 10^5$ training steps and is averaged over 5 random seeds (Kaiser et al., 2019).

## E    Further Experimental Results

### E.1    Full Evaluation Results on Atari 100K Benchmarks

| COMPONENT | ATTRIBUTE | VALUE |
|---|---|---|
| EXPLOITER | CRITIC MLP HIDDEN LAYER DIMENSIONS | $[256, 256]$ |
| | CRITIC MLP ACTIVATION FUNCTION | ReLU |
| | ACTOR MLP HIDDEN LAYER DIMENSIONS | $[256, 256]$ |
| | ACTOR MLP ACTIVATION FUNCTION | ReLU |
| | LEARNING RATE | $7.3 \times 10^{-4}$ |
| | REPLAY BUFFER SIZE | $3 \times 10^5$ |
| | BATCH SIZE | 256 |
| | NUMBER OF CRITIC ENSEMBLE | 3 |
| | DISCOUNTING FACTOR | 0.98 |
| EXPLORER | CRITIC MLP HIDDEN LAYER DIMENSIONS | $[256, 256]$ |
| | CRITIC MLP ACTIVATION FUNCTION | ReLU |
| | ACTOR MLP HIDDEN LAYER DIMENSIONS | $[256, 256]$ |
| | ACTOR MLP ACTIVATION FUNCTION | ReLU |
| | LEARNING RATE | $1 \times 10^{-4}$ |
| | REPLAY BUFFER SIZE | $3 \times 10^5$ |
| | DISCOUNTING FACTOR | 0.60 |
| SWITCHER | CRITIC MLP HIDDEN LAYER DIMENSIONS | $[64, 64]$ |
| | CRITIC MLP ACTIVATION FUNCTION | ReLU |
| | ACTOR MLP HIDDEN LAYER DIMENSIONS | $[64, 64]$ |
| | ACTOR MLP ACTIVATION FUNCTION | ReLU |
| | LEARNING RATE | $1 \times 10^{-4}$ |
| | REPLAY BUFFER SIZE | $3 \times 10^5$ |
| SEREN | SWITCHER INTERVENTION COST ($\beta$; EQUATION 5) | 0.01 |
| | NUMBER OF INITIAL EXPLORATION STEPS | 10000 |
| | FREQUENCY OF TRAINING EXPLOITER | 8 |
| | FREQUENCY OF TRAINING EXPLOITER AND SWITCHER | 4 |
| | MASK PROBABILITY FOR ENSEMBLE TRAINING | 0.2 |

Table 2: Implementation specifications for SEREN-SAC for all MuJoCo environments considered.

| Component | Attribute | Value |
|---|---|---|
| Exploiter | DQN ConvNet channels | $[32, 64, 64]$ |
| | DQN ConvNet filter size | $[8 \times 8, 4 \times 4, 3 \times 3]$ |
| | DQN ConvNet stride | $[4, 2, 1]$ |
| | DQN MLP hidden unit | $[256]$ |
| | critic MLP activation function | ReLU |
| | learning rate | $1 \times 10^{-4}$ |
| | replay buffer size | $1 \times 10^{5}$ |
| | batch size | 256 |
| | multi-step return | 10 |
| | number of critic ensemble | 3 |
| | mask probability for ensemble training | 0.2 |
| | dueling | True |
| | double Q-learning | True |
| | discounting factor $(\gamma_1)$ | 0.99 |
| Explorer | DQN ConvNet channels | $[32, 64, 64]$ |
| | DQN ConvNet filter size | $[8 \times 8, 4 \times 4, 3 \times 3]$ |
| | DQN ConvNet stride | $[4, 2, 1]$ |
| | DQN MLP hidden unit | $[256]$ |
| | critic MLP activation function | ReLU |
| | learning rate | $1 \times 10^{-4}$ |
| | replay buffer size | $1 \times 10^{5}$ |
| | batch size | 256 |
| | multi-step return | 10 |
| | number of critic ensemble | 1 |
| | dueling | True |
| | double Q-learning | True |
| | discounting factor $(\gamma_2)$ | 0.80 |
| Switcher | DQN ConvNet channels | $[32, 64, 64]$ |
| | DQN ConvNet filter size | $[8 \times 8, 4 \times 4, 3 \times 3]$ |
| | DQN ConvNet stride | $[4, 2, 1]$ |
| | DQN MLP hidden unit | $[256]$ |
| | critic MLP activation function | ReLU |
| | learning rate | $1 \times 10^{-4}$ |
| | replay buffer size | $1 \times 10^{5}$ |
| | batch size | 256 |
| | multi-step return | 10 |
| | number of critic ensemble | 3 |
| | mask probability for ensemble training | 0.2 |
| | dueling | True |
| | double Q-learning | True |
| | discounting factor $(\gamma_3)$ | 0.99 |
| SEREN | Switcher intervention cost $(\beta;$ Equation 5) | $3 \times 10^{-5}$ |
| | number of initial exploration steps | 1600 |
| | action repetitions | 4 |
| | frames stacked | 4 |
| | terminal on loss of life | True |
| | data augmentation | intensity |

Table 3: Implementation details for SEREN-Eff-DQN for all games in the Atari 100K benchmarks.

| Games | Rainbow | Efficient Rainbow | Efficient DQN | DrQ | SEREN-DrQ |
|---|---|---|---|---|---|
| Alien | 318.7 | **739.9** | 558.1 | 702.5 | 721.8 |
| Amidar | 32.5 | **188.6** | 63.7 | 100.2 | 173.4 |
| Assault | 231.0 | 431.2 | 589.5 | 490.3 | **596.8** |
| Asterix | 243.6 | 470.8 | 341.9 | **577.9** | 436.1 |
| BankHeist | 15.6 | 51.0 | 74.0 | 205.3 | 145.8 |
| BattleZone | 2360.0 | **10124.6** | 4760.8 | 6240.0 | 5287.3 |
| Boxing | -24.8 | 0.2 | -1.8 | 5.1 | **5.8** |
| Breakout | 1.2 | 1.9 | 7.3 | 14.3 | **14.6** |
| ChopperCommand | 120.0 | 861.8 | 624.4 | 870.1 | **1474.5** |
| CrazyClimber | 2254.5 | 16195.3 | 5430.6 | **20072.2** | 16679.1 |
| DemonAttack | 163.6 | 508.0 | 403.5 | **1086.0** | 907.2 |
| Freeway | 0.0 | **27.9** | 3.7 | 20.0 | 26.9 |
| FrostBite | 60.2 | 866.8 | 202.9 | **889.9** | 761.5 |
| Gopher | 431.2 | 349.5 | 320.8 | 678.0 | **707.9** |
| Hero | 487.0 | **6857.0** | 2200.1 | 4083.7 | 4495.5 |
| JamesBond | 47.4 | 301.6 | 133.2 | **330.3** | 284.5 |
| Kangaroo | 0.0 | 779.3 | 448.6 | **1282.6** | 921.0 |
| Krull | 1468.0 | 2851.5 | 2999.0 | 4163.0 | **4443.1** |
| KungFuMaster | 0.0 | **14346.1** | 2020.9 | 7649.0 | 8937.5 |
| MsPacman | 67.0 | 1204.1 | 872.0 | 1015.9 | **1396.9** |
| Pong | -20.6 | -19.3 | -19.4 | -17.1 | **−16.3** |
| PrivateEye | 0.0 | 97.8 | **351.3** | -50.4 | 100.0 |
| Qbert | 123.5 | **1152.9** | 627.5 | 769.1 | 1043.3 |
| RoadRunner | 1588.5 | **9600.0** | 1491.9 | 8296.3 | 7835.0 |
| Seaquest | 131.7 | 354.1 | 240.1 | 299.4 | **395.9** |
| UpNDown | 504.6 | 2877.4 | 2901.7 | 2901.7 | **4534.2** |
| Median Human Normalised Score | 0.020 | 0.194 | 0.064 | 0.247 | **0.253** |
| Mean Human Normalised Score | 0.045 | 0.290 | 0.144 | 0.377 | **0.396** |

Table 4: Evaluation on Atari 100K Benchmarks.

