# OpenReview forum: "Systematic Exploration and Exploitation via a Markov Game with Impulse Control"
_TMLR — Rejected by TMLR_

### Review · Reviewer_xtTx · 2024-04-09

**Summary Of Contributions:**

This paper approaches the exploration-exploitation problem in reinforcement learning and proposes to frame the issue as a game between an exploiter agent and a switcher agent that can switch between the exploiter and a third explorer agent (learned separately). In the proposed SEREN algorithm, the exploration agent maximizes an ensemble-based uncertainty measure, the exploiter maximizes the task reward, and the switcher reduces system uncertainty subject to some switching cost. Some theory is presented that the method converges in a linear case and experiments are performed in gridworld, mujoco, and atari tasks.

**Audience:**

No

**Claims And Evidence:**

No

**Requested Changes:**

Each of the weaknesses raised above would need to be addressed. Issue 1 covers related work that is relevant to both theory and experiments. Issues 2 and 3 cover issues with the theory to be addressed. Issues 4, 5, 6, and 7 cover issues with the experiments that need to be addressed. I would also like to see a more clear discussion of the question raised above.

**Strengths And Weaknesses:**

### Strengths
1. The paper approaches an important problem from a new perspective. Exploration is an important issue and this paper tries to bring in ideas about games and impulse control to replace the standard notion of adding exploration bonuses or noise to a single policy. This is to my knowledge a novel idea and potentially interesting.

2. The algorithm is clearly presented and the paper is generally well-written.

3. It is nice that the SEREN method can be flexibly applied on top of different existing base methods and/or uncertainty quantifiers.

### Weaknesses
1. The paper ignores related work on principled exploration and the standard regret formulation of this problem. The standard way to approach the exploration-exploitation tradeoff in theory dating back to at least [1] is via regret bounds that give explicit rates on how many suboptimal (i.e. exploration) steps are necessary until an optimal (i.e. exploit) policy can be found. This has been extended to linear function approximation setting for UCB [2,3] and even to infinite dimensional features (e.g. kernels) [4], nonlinear functions [5] and other generalizations [6,7]. While the approaches listed above are primarily on the theory side, another line of work attempts to bridge theory and practice to solve hard exploration problems in principled ways. For example [8] uses a PG-base method and excels at hard exploration problems and [9,10,11,12] use value-based methods inspired by theory and applied on hard exploration tasks.

2. The theory does not make the assumptions clear in the paper and upon reading the assumptions in the appendix, they are too strong for the problem at hand. In particular, the theory assumes that the MDP is ergodic. This is a standard assumption for *value learning* theory since it essentially assumes away the exploration problem. For exploration theory, this sort of assumption assumes away the actual issue, which is why we usually use the regret formulation.

3. The theory does not give any clear bound on the rate at which it converges, which is important in these problems. In general, even simple exploration methods can converge to the optimal policy if given enough time since a random policy eventually reaches all states, but the important question is how fast this happens.

4. The paper lacks relevant baseline methods in the experiments. In particular, the method is compared against baselines that are not explicitly attempting to do smart exploration strategies like DQN and SAC, and thus it is difficult to assess whether SEREN is a good exploration strategy or whether it is just good to add any exploration strategy and SEREN happens to be one.

5. The experiments to not directly study exploration. The tasks considered: simple gridworld, mujoco, and atar are not particularly hard exploration problems (except for perhaps a few of the atari games within the total benchmark). This makes it difficult to assess whether the proposed method is actually improving exploration or helping with other issues or just using more hyperparameter tuning.

6. The empirical improvements are not very clear. Even looking at the results of the experiments on not particularly well-chosen tasks, the improvements over the baselines are not very clear over SAC and DrQ, which do not focus on exploration. There is a clear gain in the gridworld, but this is not so surprising as standard DQN has many known issues with exploration.

7. The method introduces quite a few extra hyperparameters and does not discuss how they are tuned. The appendix lists final hyperparameter values, but when so many hyperparameters are introduced (additional exploration and switcher agents with all of their parameters, intervention costs, and warmup period), it is possible that with enough extra tuning relative to the baseline methods, a better result could be found on some particular benchmarks. These extra hyperparameters are a practical issue for the method, but also without knowing how they are tuned it makes the validity of the results unclear.

### Questions
1. The three agents together effectively define one actual policy that gets implemented in the environment. Can this policy be seen as maximizing some combination of their objectives? Is this meaningfully different from a single agent that optimizes the objectives jointly?

### Summary
Overall, I think this paper needs substantial work and re-framing before resubmission.


### References

[1] Auer, P., Jaksch, T., & Ortner, R. (2008). Near-optimal regret bounds for reinforcement learning. Advances in neural information processing systems, 21.

[2] Jin, C., Yang, Z., Wang, Z., & Jordan, M. I. (2020, July). Provably efficient reinforcement learning with linear function approximation. In Conference on learning theory (pp. 2137-2143). PMLR.

[3] Zanette, A., Brandfonbrener, D., Brunskill, E., Pirotta, M., & Lazaric, A. (2020, June). Frequentist regret bounds for randomized least-squares value iteration. In International Conference on Artificial Intelligence and Statistics (pp. 1954-1964). PMLR.

[4] Du, S., Kakade, S., Lee, J., Lovett, S., Mahajan, G., Sun, W., & Wang, R. (2021, July). Bilinear classes: A structural framework for provable generalization in rl. In International Conference on Machine Learning (pp. 2826-2836). PMLR.

[5] Agarwal, A., Jin, Y., & Zhang, T. (2023, July). VO $ Q $ L: Towards Optimal Regret in Model-free RL with Nonlinear Function Approximation. In The Thirty Sixth Annual Conference on Learning Theory (pp. 987-1063). PMLR.

[6] Jin, C., Liu, Q., & Miryoosefi, S. (2021). Bellman eluder dimension: New rich classes of rl problems, and sample-efficient algorithms. Advances in neural information processing systems, 34, 13406-13418.

[7] Foster, D. J., Kakade, S. M., Qian, J., & Rakhlin, A. (2021). The statistical complexity of interactive decision making. arXiv preprint arXiv:2112.13487.

[8] Agarwal, A., Henaff, M., Kakade, S., & Sun, W. (2020). Pc-pg: Policy cover directed exploration for provable policy gradient learning. Advances in neural information processing systems, 33, 13399-13412.

[9] Ren, T., Zhang, T., Szepesvári, C., & Dai, B. (2022, August). A free lunch from the noise: Provable and practical exploration for representation learning. In Uncertainty in Artificial Intelligence (pp. 1686-1696). PMLR.

[10] Zhang, T., Ren, T., Yang, M., Gonzalez, J., Schuurmans, D., & Dai, B. (2022, June). Making linear mdps practical via contrastive representation learning. In International Conference on Machine Learning (pp. 26447-26466). PMLR.

[11] Henaff, M., Raileanu, R., Jiang, M., & Rocktäschel, T. (2022). Exploration via elliptical episodic bonuses. Advances in Neural Information Processing Systems, 35, 37631-37646.

[12] Misra, D., Henaff, M., Krishnamurthy, A., & Langford, J. (2020, November). Kinematic state abstraction and provably efficient rich-observation reinforcement learning. In International conference on machine learning (pp. 6961-6971). PMLR.

---

### Review · Reviewer_nBQy · 2024-04-20

**Summary Of Contributions:**

The paper introduces the Selective Reinforcement Exploration Network (SEREN), a method for addressing the exploration-exploitation trade-off in Reinforcement Learning (RL) by using a dual-agent setup with one agent focusing exclusively on exploitation (Exploiter), maximizing task-dependent rewards, and the other (Switcher) controls when to shift to an exploration mode to minimize system uncertainty. The key part of the method is the impulse control policies, which allow the Switcher to determine the timing and states for switching between exploration and exploitation. The authors prove the convergence of SEREN in a linear setting and demonstrate its effectiveness through experiments on simple RL environments.

**Audience:**

Yes

**Claims And Evidence:**

No

**Requested Changes:**

I strongly believe that this paper could make for a very good submission to any top tier ML venues *if* the experimental section was significantly improved. Of the weaknesses above, I think it should be relatively straightforward to make progress on W1 by including more minigrid / babyAI environments. That would not be sufficient to pass the bar, and I would like to see some degree of improvement also wrt. W2 and W3 (in that order).

General nits:

- There are few instances of a bunch of words that inconsistently used in the manuscript (e.g. trade-off vs tradeoff). The manuscript could be tidied up.
- The extra hyperparams that come with SEREN are not free, and I would like to see some discussion on how to select discount factor and generally tune the impulse control policies for new problems (e.g. what are common pitfalls to avoid?)
- I think the paper could use some discussion on (a) scaling wrt. sparsity of reward signal, and (b) scaling wrt. state space and action space dimensionality.

**Strengths And Weaknesses:**

** Strenghts

[S1]  I like the presented approach. Using multi-agent frameworks for figuring out better ways to explore at learning time is a great and fun idea, and it's not a common setup found in the literature that I think the community should explore seriously.

[S2] The authors have done a good job of describing the method, and the theoretical justifications seem sound (particulary wrt. appendix B). Section 3 and 4 are clear, and I reckon it wouldn't be too hard to reproduce the method starting from the manuscript.

[S3] It is great that SEREN can be applied to many different RL baselines relatively cleanly, and the authors have done a good job at showing this experimentally.

** Weaknesses

The primary, and major, weakness of this work is its empirical evaluation.

[W1] The environments chosen have historically been used for exploration papers, but the world of RL has moved on and new environments that provide better problems for benchmarking improvements on exploration do exist (e.g. NetHack / MiniHack, MineRL, procgen, etc). The choice of environments in the manuscript makes it really hard to qualitatively say whether SEREN beats the baselines. Furthermore, MiniGrid includes a pletora of environments within BabyAI that would make for a fantastic set of (cheap!) experiments, and it's not clear to me why only the simplest possible version of a minigrid problem was employed.

[W2] This manuscript is clearly a "method" paper (rather than a "problem" paper), and there's a vast literature of work looking at improving exploration methods. The baselines used in the paper, while decent for a single-agent RL work, are poorly chosen. The manuscript misses comparisons against not just methods that do standard epsilon-greedy-like exploration, but those that primarily focus on solving the exploration problem directly (e.g. Random Network Distillation and follow-up work).

[W3] The experiments seem to all be about sample efficiency and performance, but there are many things that one would want to know when evaluating new exploration systems. See e.g. the relatively simple evaluation made in https://arxiv.org/pdf/2108.11811.pdf

Overall, this makes acceptance of this manuscript as it stands really difficult, because it puts the burden of evaluating the effectiveness of SEREN nearly entirely on the community.

---

### Review · Reviewer_L6sr · 2024-04-21

**Summary Of Contributions:**

The paper focuses on the problem of managing the exploration-exploitation tradeoff in single agent reinforcement learning. The key idea in this work is to separate the roles of exploiting known rewards and exploring novel states (thereby reducing state space uncertainty), and delegate each role to a separate agent, namely, Exploiter and Switcher. To this end, the paper introduces the Selective Reinforcement Exploration Network (SEREN), that instantiates both the Exploiter and the Switcher as RL agents and manages the exploration-exploitation tradeoff by inducing a Markov game between these two agents. The Switcher uses an impulse control policy to decide when to switch to an exploration mode and in what state, effectively minimizing system uncertainty while the exploiter focuses on optimizing for the task reward on all other states. Authors claim that such a method results in a plug-and -play-approach that can be seamlessly augmented with any existing RL algorithm which they empirically demonstrate using DQN and SAC as baseline algorithms. The paper provides convergence results for agent’s value functions. Further, experiments on three different environments (MiniGrid, MuJoCo an Atari) have been conducted to assess the efficacy of the proposed approach in terms of overall performance and ablation studies.

**Audience:**

Yes

**Broader Impact Concerns:**

This method provides a potential of explicit control over intervention times via hyper-parameters and an explicit optimization over state visitation. While the current scope of the work does not warrant investigation into broad impacts, it would be useful for such decoupled approaches to discuss/think about impact when deployed for real-world systems. One such aspect for this work would related to fairness and bias, namely, given that SEREN learns to differentiate between exploitation and exploration, how can one ensure that the system does not develop biased strategies towards certain states or decisions?

**Claims And Evidence:**

No

**Requested Changes:**

- Overall, the writing of the paper needs improvement and proofreading as there are some incoherent and in accessible part of the manuscript. (Minor: the last paragraph in Conclusion section has some construction error and not clear what it states.)
- It would be very helpful if the authors provide detailed methodological justification over existing methods. Further, the authors need to explain the role of switcher, corresponding architecture and Equation 5 better.
- It is important to include environments with larger or more complex state space to understand the efficacy of the approach. It is also required to showcase the plug and play approach and improvements with at least PPO and A3C algorithms in addition to the one provided currently.
- As this method explicitly focuses on decreasing uncertainty of state space, it needs to provide an in-depth experimental study for the same. For instance, reward curves provide the information about overall policy. But it is recommended that authors consider providing the performance tradeoff vs. the no of states visited. It would also be helpful to include analysis on state space, state coverage and density achieved by various approaches and visualization or statistics related to uncertainty decreasing over the state space.
- In the experiments, it is important to discuss and analyze both $\gamma$ and $\beta$ and provide guidance on how practitioners should think abouth these hyper-parameters.
- I understand that this is a tricky one but there are lot of exploration strategies available and without showing empirical comparisons to them on a suite of complex environments where this approach performs significantly better, it is very hard to say anything about the efficacy of this approach. This is especially true given the complexity such an approach would induce as environments become more complex.
- There is a highly related work on similar lines for control [1] which needs to be discussed and potentially compared.

[1] Dual Control of Exploration and Exploitation for Auto-Optimisation Control with Active Learning, IEEE 2024

**Strengths And Weaknesses:**

Strengths:
------------

- The exploration-exploitation tradeoff is a classical problem in reinforcement learning which is still an open challenge and new contributions in this space are valuable.
- The key idea of casting this exploration-exploitation tradeoff into a Markov game, thereby adopting a separation of concerns approach is novel with significant potential implications.
- The method automatically learns both aspects - which state to trigger the switching mechanism at and when to make the switch which is a plus.
- Theoretical analysis regarding the convergence of value function of agents is sound and insightful.
- Experiments with DQN and SAC provide useful insights about the effectiveness of plug-and-play approach.
- The results for DQN on min-grid environment are strong.


Weaknesses:
---------------

- While the idea of managing exploration and exploitation separately is novel, it is not clear how scalable or effective it is to consider them as different tasks being optimized by two different agents. The Markov game induction introduces extra complexities of non-stationarity and credit assignment into the problem which is already complex in the single agent version. The theoretical justification of convergence mitigates these concerns slightly but it is not adequate to support eh general applicability of the method, given they are specifically shown to hold only in linear regime. Further, the implications of using such approach in practice are far reaching (and probably difficult to cover/anticipate) especially when the state place is complex (discussed later).
-  There exists a significant amount of efforts towards addressing this problem and the paper does a good job in covering them in their related work. However, from the descriptions, it is not clear how the existing approaches are inferior to the proposed approach. While the authors argue that the proposed approach provides for improved training, this is not really justified through experiments.
- Some parts of the method and algorithm requires improvement in presentation and explanation of some design choices:
    - Authors mention that the decision to switch from exploitative behavior to exploratory behavior is learned which I follow but they also state that the vice versa is also learned. It is not clear how the method learns to go back from exploratory to exploitative behavior.
    - Fig 1 is a bit confusing. Does the exploiter and explorer have the same policy network/shared parameters? Does Fig 1 mean that the same policy behave as explorer when $g=0$ and exploiter when $g=1$?
    - Minor: why do you choose .it and .re to signify Exploiter and Switcher? They don’t match directly.
    - One of the surprising parts of the approach is the evaluation of both $R_t^{it}$and $R_t^{re}$ at all time steps. It is not clear why this aligns with the independent management of exploration and exploitation. Crucially, it is possible that joint system using both rewards together amounts to not a very different outcome than regularization approaches where the explorer/switcher optimization is regularizing the exploiter optimization and then such a separation seems like an overkill.
    - The choice of using model-free instantiation of the SEREN approach is not justified so is not the use of ensemble approach. Specifically, the use of $\beta$ is very arbitrary and could turn out to be a very sensitive hyper-parameter. As a follow-up, it is hard to discern the details of Eq 5 and requires elaboration and more accessible explanation.
    - Algorithm 1 seems to be using off-policy data for updating the two agents. Is this correct?  If yes, how does the approach deals with stale data and any required corrections?
- In addition to the concerns with the methodology, the major weakness of this work stems from its experiments:
    - The environments used in the paper are quite simple and fail to test the efficacy of the approach effectively. It is important to use more complex environments with large state space and where exploration is a combinatorial problem difficult to solve.
    - Similarly, while it is good to see the use of plug and play approach with DQN and SAC, the authors did not include other state-of-the-art RL methods that have their own exploration strategies, such as PPO or A3C. As these methods are widely used currently in many systems, this is a big miss.
    - Contrary to authors claims, Figure 3 does not clearly depict the robustness of the current approach as SEREN does have a lot of variance any improvement seen here is marginal which could have been accounted by tuning of other methods.
    - Ablation of $\gamma_{re}$ is nice to see although the numbers considered are 0.1, 0.6 and 0.98. It is not clear why the authors consider these values as too small or large. Table 1 just shows that the approach is highly sensitive to this hyper-parameter and even a small change can significantly affect the performance.
    - Figure 4 is a bit strange. It is correct that we expect the switcher to stop active participation as the system uncertainty becomes lower. However, the two provided examples have very different issues: Comparing Figure 3 and 4, In the Ant environment, it appears that switching completely dies down before the agents start to learn something useful (i.e. they hit a performance where they become somewhat comparable and they still have lot of learning left). So any improvement here could be because switcher helped to put the agents into slightly better states (good initialization/direction of learning process) but this is not explained or not easy to discern. For Reacher environment, the agents have learned pretty fast for all methods but the switching keep happening a lot for the entire horizon (the variance is fairly high in switching actions graph).

---

### Decision · Action_Editor_7BSN · 2024-07-09

**Recommendation:** Reject

**Comment:**

The reviewers found several strengths in this paper: nicely casting exploration-exploitation tradeoff as a Markov game; automatically learning a triggering mechanism to switch-on an exploration policy; offering theoretical analysis on the convergence of the value function of the exploiter-explorer agents; and reporting strong DQN results on the min-grid environment.

However, the weaknesses were also unanimously recognized - the prime concern being weak experimental results lacking baseline comparisons to other exploration strategies; and stress testing the methods on harder problems with larger state spaces. Quality of writing and clarity of motivation, theoretical assumptions and design choices were also raised as a concern.

Unfortunately, the authors did not engage during the discussion and hence the reviewers could not be moved past their initial evaluation.

**Audience:**

Exploration-exploitation in RL is an evergreen problem. As such there is an audience for this topic. However, without accurate & convincing claims, the findings may not be reliable.

**Claims And Evidence:**

The claims made in the submission are NOT supported adequately by accurate, convincing and clear evidence. Here are the reasons documented in the reviews:

- The writing of the paper needs improvement with edits to some incoherent sections
- The motivation of the proposed architecture, the assumptions needed for theory to work, are not fully clear.
- To show-case the claimed plug-and-play feature, benchmarking on problems involving larger & more complex state spaces would make the contribution more convincing.
- More insights into  state coverage and density achieved by various approaches should have been included.
- Overall a consistent theme in the reviews was weak experimentation:  without showing significantly favorable empirical comparisons to exploration strategies proposed in the literature on a suite of complex environments, it is very hard to reach a conclusion on the claimed efficacy of this method.